# Protein sorting by lipid phase-like domains supports emergent signaling function in B lymphocyte plasma membranes

Matthew B Stone[†], Sarah A Shelby[†], Marcos F Núñez, Kathleen Wisser, Sarah L Veatch*

Department of Biophysics, University of Michigan, Ann Arbor, United States

**Abstract** Diverse cellular signaling events, including B cell receptor (BCR) activation, are hypothesized to be facilitated by domains enriched in specific plasma membrane lipids and proteins that resemble liquid-ordered phase-separated domains in model membranes. This concept remains controversial and lacks direct experimental support in intact cells. Here, we visualize ordered and disordered domains in mouse B lymphoma cell membranes using super-resolution fluorescence localization microscopy, demonstrate that clustered BCR resides within ordered phase-like domains capable of sorting key regulators of BCR activation, and present a minimal, predictive model where clustering receptors leads to their collective activation by stabilizing an extended ordered domain. These results provide evidence for the role of membrane domains in BCR signaling and a plausible mechanism of BCR activation via receptor clustering that could be generalized to other signaling pathways. Overall, these studies demonstrate that lipid mediated forces can bias biochemical networks in ways that broadly impact signal transduction.

*For correspondence: sveatch@umich.edu

[†]These authors contributed equally to this work

Competing interests: The authors declare that no competing interests exist.

## Introduction

Cells interact with their environment through a complex set of biochemical networks that transmit information across the plasma membrane. Often, signal transduction relies on the spatial organization of receptors as well as effector proteins that regulate down-stream signaling activity. In principle, spatial organization in biological membranes can be enforced through varied mechanisms including direct protein-protein interactions (*Douglass and Vale, 2005*; *Su et al., 2016b*), dynamic or passive coupling to cytoskeletal elements (*Wülfing and Davis, 1998*; *Kaizuka et al., 2007*; *DeMond et al., 2008*), adhesion (*Davis and van der Merwe, 2006*), curvature mediated forces (*Zhu et al., 2012*; *Aimon et al., 2014*), or steady state biochemical networks with spatial heterogeneity (*Chau et al., 2012*). It is also proposed that plasma membrane lipids contribute to the spatial organization of membrane proteins via the same thermodynamic forces that drive the separation of liquid-ordered and liquid-disordered phases in model membranes (*Schroeder et al., 1994*; *Lingwood and Simons, 2010*). Liquid-ordered like domains are often referred to as lipid rafts or lipid shells (*Simons and Ikonen, 1997*; *Anderson and Jacobson, 2002*), and are hypothesized to impact a broad array of signaling cascades that originate at the plasma membrane (*Simons and Toomre, 2000*) including B cell receptor signaling (*Cheng et al., 1999*). However, the existence of phase-like membrane domains and their putative roles in signaling pathways remain controversial, largely because the majority of experimental support for this concept is indirect or relies on methodology with well characterized limitations (*Heerklotz, 2002*; *Munro, 2003*; *Kwik et al., 2003*; *Kenworthy, 2008*).

**eLife digest** Membranes made of molecules called lipids surround every living cell to protect the cell's contents. Cells also communicate with the outside environment via their membranes. Proteins in the membrane receive information from the environment and trigger signaling pathways inside the cell to relay this information to the center of cell. The way in which proteins are organized on the membrane has a major influence on their signaling activity.

Some areas of the membrane are more crowded with certain lipids and signaling proteins than others. Lipid and protein molecules of particular types can come together and form distinct areas called "ordered" and "disordered" domains. The lipids in ordered domains are more tightly packed than disordered domains and it is thought that this difference allows domains to selectively exclude or include certain proteins. Ordered domains are also known as "lipid rafts". Lipid rafts and disordered domains may help cells to control the activities of signaling pathways, however, technical limitations have made it difficult to study the roles of these domains.

The membranes surrounding immune cells called B cells contain a protein called the B cell receptor, which engages with proteins from microbes and other foreign invaders. When the B cell receptor binds to a foreign protein it forms clusters with other B cell receptors and becomes active, triggering a signaling pathway that leads to immune responses.

Stone, Shelby et al. examined lipid rafts and disordered domains in B cells from mice using a technique called super-resolution fluorescence microscopy. The results show that clusters of B cell receptors are present within lipid rafts. These clusters made the lipid rafts larger and more stable. A protein that is needed during the early stages of B cell receptor signaling was also found in the same lipid rafts. Another protein that terminates signaling was excluded because it prefers disordered domains. Together, this provides a local environment in certain areas of the membrane that favors receptor activity and supports the subsequent immune response.

Future work is needed to understand how cells control the make-up of lipids and proteins within their membranes, and how defects in this regulation can alter signaling activity and lead to disease.

Some of the strongest experimental evidence supporting a heterogeneous plasma membrane comes from Förster resonance energy transfer (FRET) measurements between membrane components. This method is sensitive to heterogeneity on length-scales smaller than the Förster distance (~5 nm) (*Kenworthy and Edidin, 1998*; *Varma and Mayor, 1998*; *Pyenta et al., 2001*; *Zacharias et al., 2002*; *Sharma et al., 2004*; *Rao and Mayor, 2005*; *Sengupta et al., 2007*; *Goswami et al., 2008*). While powerful for detecting interactions between proteins and/or lipids that occur on small length-scales, the FRET signal is highly nonlinear with respect to probe separation distance and depends on both donor/acceptor ratio and probe absolute density, which typically vary in experiments. These complications can lead to weak sensitivity of FRET measurements to changes in local concentration and often modeling is required to interpret experimental findings quantitatively (*Kenworthy and Edidin, 1998*; *Rao and Mayor, 2005*). Here, we apply super-resolution fluorescence localization imaging to complement these past approaches to directly visualize ordered and disordered-like domains in intact cell plasma membranes. This approach allows us to characterize and quantify, in a model-independent manner, the spatial organization of membrane components on length scales between those accessible by FRET-based techniques and conventional optical microscopy.

B cells undergo a signaling response when their B cell receptors (BCRs) are engaged by antigen, either in the form of solution-phase multivalent antigen (*Minguet et al., 2010*) or surface-presented monovalent antigen (*Batista et al., 2001*), however the molecular mechanisms driving BCR signal initiation are still controversial (*Packard and Cambier, 2013*). Notably, simply clustering the BCR with antibodies directed against the receptor initiates phosphorylation by Src family kinases, resulting in the binding of downstream kinases and effectors that amplify and propagate the signaling response (*Cambier and Ransom, 1987*; *Campbell and Sefton, 1990*). This supports the idea that, at least in some contexts, the spatial organization of the BCR can function to communicate antigen binding, similar to other transmembrane receptors (*Metzger, 1992*), as opposed to a mechanism where

receptor binding is conveyed solely through ligand-induced conformational changes (*Campbell and Humphries, 2011*; *Sounier et al., 2015*). Several mechanistic models have been put forth to explain how BCR clustering could give rise to receptor activation. In one model, BCR clustering initiates signaling via protein-protein interactions between neighboring receptors, such as the transphosphorylation of nearby receptors by receptor-bound kinases (*Sotirellis et al., 1995*; *Kurosaki and Kurosaki, 1997*). A second model proposes that antigen binding acts to separate pre-clustered BCR, exposing binding sites to kinases that propagate activation (*Yang and Reth, 2010a*, *2010b*). A third model, which is the focus of investigation here, postulates that clustering BCR acts to stabilize an ordered membrane domain that impacts the receptor-proximal distribution of regulatory proteins involved in initiating or modulating the resulting cellular response. Specifically, ordered membrane domains are predicted to support interactions with activating kinases and suppress interactions with deactivating phosphatases. This hypothesis has been strengthened by experimental support (*Pierce, 2002*; *Sohn et al., 2006*; *Gupta and DeFranco, 2007*; *Sohn et al., 2008a*), and among this evidence are observations of changes in FRET upon receptor clustering and activation between BCR and a marker of ordered membrane domains, but not between BCR and a marker of disordered domains (*Sohn et al., 2006*, *2008b*). These studies demonstrated that receptor clustering leads to a transient increase in near-neighbor interactions (within a few lipid diameters) between BCR and order-preferring lipid probes and suggested that these interactions are important for the initiation of BCR activation by the Src family kinase Lyn. Currently, the role of membrane domains in BCR signaling remains a topic of active investigation (*Pierce and Liu, 2010*; *Horejsi and Hrdinka, 2014*) as the field works to put together a more holistic picture of how interactions between proteins and lipids could support signaling function.

Here, we use super-resolution fluorescence localization microscopy to characterize the lipid environment around BCR clusters. Using this approach, we directly visualize ordered domains co-localized with clustered BCR. We find that these domains sort key regulatory proteins involved in BCR signaling and provide a local environment that favors tyrosine phosphorylation. We also present a predictive model whereby BCR clustering leads to its phosphorylation through the stabilization of an ordered membrane domain. Our findings suggest that the collective protein-lipid and lipid-lipid interactions responsible for the stabilization of an ordered domain around BCR clusters give rise to emergent signaling function by influencing the local biochemical environment of BCRs within clusters. These measurements detail the molecular redistribution of membrane components around embedded membrane protein clusters, utilizing super-resolution microscopy to gain access to length scales that are smaller than those accessible by conventional microscopy and larger than length scales accessible by FRET. Overall, our imaging studies provide additional direct evidence that clustered BCR associates with a local environment enriched in ordered lipids. Together with our simulation and functional data, this evidence supports a mechanism for clustering-induced activation of BCR that involves lipid-mediated sorting of regulatory proteins. Similar mechanisms may be relevant to other pathways where changes in receptor organization lead to signaling functions.

## Results

### Phase-like domains observed in intact B cells

Membrane heterogeneity was probed using established markers of ordered and disordered membrane domains (*Figure 1a* and *Figure 1—figure supplement 1*). Disordered domains were marked with a short transmembrane peptide (TM) and a peptide anchored to the inner leaflet through a polybasic sequence and geranylgeranyl modification (GG) (*Pyenta et al., 2001*; *Baumgart et al., 2007*; *Leventhal et al., 2010*). Ordered membrane domains were marked with a minimal lipidated peptide anchored to the inner leaflet through saturated palmitoyl and myristol modifications (PM) (*Pyenta et al., 2001*; *Baumgart et al., 2007*), or with cholera toxin subunit B (CTxB), which binds to the ganglioside GM1 on the outer leaflet of the plasma membrane. Probe partitioning was verified in isolated giant plasma membrane vesicles (GPMVs) imaged at low temperature as demonstrated in *Figure 1—figure supplement 2*. PM, TM, and GG all lack specific protein interaction domains; therefore their spatial distributions are determined by their interactions with the plasma membrane.

We directly observed the sorting of peptides to and away from protein clusters in cell membranes using multi-color super-resolution fluorescence localization imaging (*Betzig et al., 2006*; *Hess et al.,*

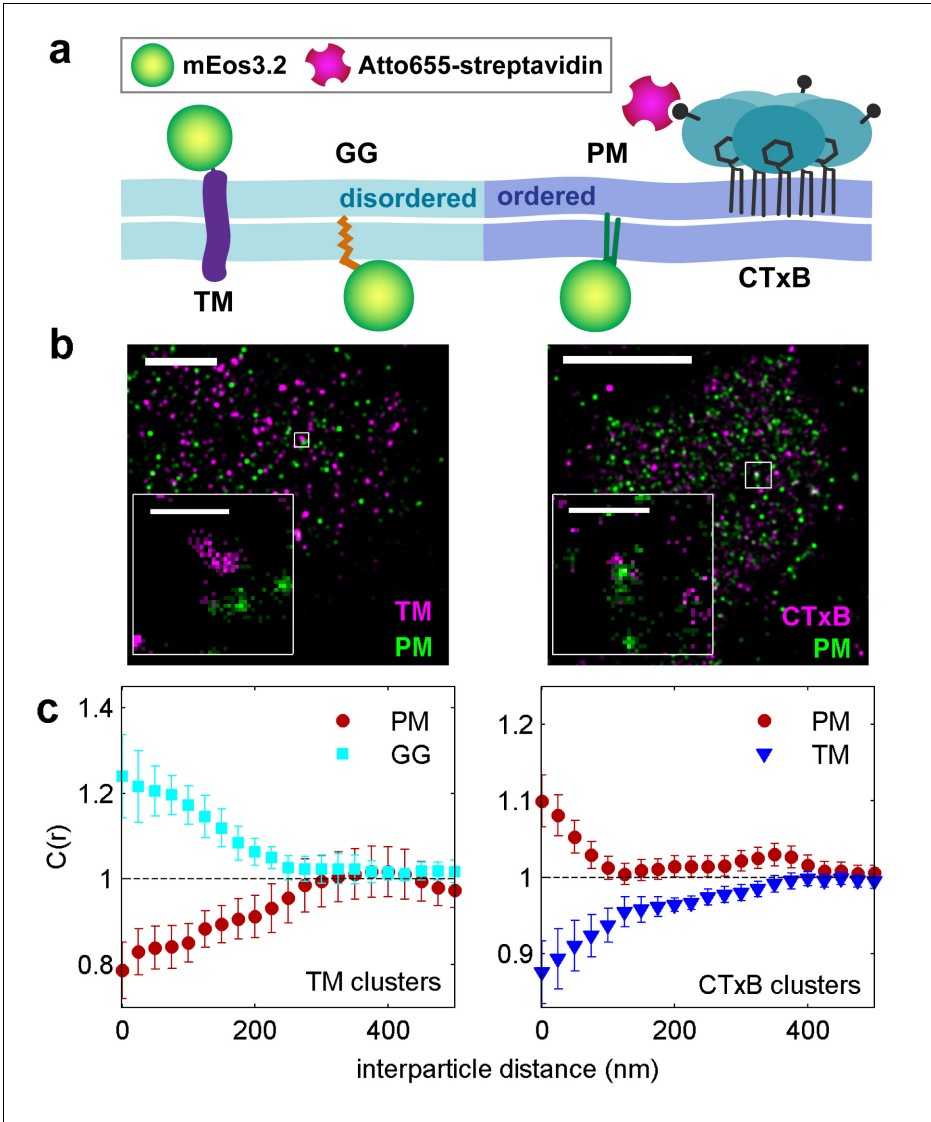

**Figure 1.** Clusters of ordered or disordered phase markers create distinct membrane domains. (a) Schematic representation of minimal anchor peptides and their phase preference as determined from model membranes. Amino acid sequences and chemical structures are shown in *Figure 1—figure supplement 1*. (b) Reconstructed super-resolution images of PM with either clustered TM (left) or clustered CTxB (right). Scale-bars are 5 µm and 500 nm in the inset. (c) Cross-correlation functions, C(r), of PM and GG constructs in cells containing clustered TM (left) or of PM and TM constructs in cells containing clustered CTxB (right). A value of C(r) = 1 indicates a random co-distribution, C(r) > 1 indicates co-clustering, and C(r) < 1 indicates exclusion. These curves represent an average over multiple individual cells from multiple experiments. Curves are averaged over the following number of cells: TM with PM (12) or GG (8), and CTxB with PM (58) or TM (27). Error-bars indicate the SEM between cells. Curves for individual cells are shown in *Figure 1—figure supplement 4*. Additional representative images for all conditions are shown in *Figure 1—figure supplement 7*.

The following figure supplements are available for figure 1:

**Figure supplement 1.** Amino acid sequences of membrane anchors used in this study.

**Figure supplement 2.** Membrane anchors partition into different phases in GPMVs.

**Figure supplement 3.** Finite lateral resolution and incomplete spatial sampling impacts measured cross-correlations.

*Figure 1 continued on next page*

*Figure 1 continued*

**Figure supplement 4.** Correlation functions from individual cells and average curves.
**Figure supplement 5.** Distribution of correlation function values closely matches the width expected from single measurement errors.
**Figure supplement 6.** The cross-correlation amplitude is weakly dependent on lipid probe expression level.
**Figure supplement 7.** Representative images from *Figure 1*.

*2006*; *Heilemann et al., 2009*) and a quantitative cross-correlation analysis (*Sengupta et al., 2011*; *Veatch et al., 2012*; *Stone and Veatch, 2015*) following methods detailed in Materials and methods. First, we stabilized a disordered domain in the plasma membrane by clustering transiently expressed TM through the binding of primary and secondary antibodies, then imaged these clusters in combination with transiently expressed PM or GG peptides (*Figure 1b*, left panel). Pair cross-correlation functions indicate that TM clusters are enriched in GG and depleted of PM on average compared to the membrane as a whole (*Figure 1c*, left panel), consistent with the partitioning of these probes into disordered domains in GPMVs. In separate experiments we stabilized an ordered domain by clustering biotinylated CTxB bound to endogenous GM1 with streptavidin, and imaged these clusters in combination with PM or TM peptides (*Figure 1b*, right panel). Pair cross-correlation functions indicate that PM is enriched and TM is depleted on average within CTxB clusters (*Figure 1c*, right panel), in good agreement with the partitioning of these probes into ordered domains in GPMVs.

The spatial heterogeneity observed within TM or CTxB clusters is subtle; we observed at most 20% enrichment or depletion of the peptide probes within the clustered protein domains, although this is an underestimate of the actual enrichment or depletion due to the finite lateral resolution of the measurement (*Figure 1—figure supplement 3*). Relevant to this point, we found that TM clusters (68 nm) were on average larger than CTxB clusters (40 nm), which were on the order of our cumulative resolution (40 nm) in the measurements summarized in *Figure 1c*. It is possible that the weaker enrichment and depletion of lipid probes within CTxB clusters as compared to TM clusters is simply due to the limited resolution of these measurements.

Cross-correlation curves are aggregated from multiple (8-58) single-cell measurements, and error bars on *Figure 1c* indicate the standard error of the mean (SEM) between curves generated from single cells (*Figure 1—figure supplement 4*). Error in these measurements was dominated by probe sampling statistics (*Figure 1—figure supplement 5*) because the surface density of probes (2–20 $\mu m^{-2}$) is such that only a few peptides co-localize with nanosized clustered protein domains within an image of a single chemically fixed cell. We note that the apparent surface area occupied by fluorescent peptides is likely a vast under-estimate of the surface area occupied by ordered or disordered phase-like domains since these probes are only one of many membrane components that likely occupy these domains, leading to inherent under-sampling of space (*Figure 1—figure supplement 3*) (*Stone et al., 2017*). In addition, incomplete spatial sampling by fluorescent peptides can give rise to the appearance of peptide self-clustering, since a single mEos3.2 fluorophore can reversibly photo-switch (*Endesfelder et al., 2011*) and therefore is likely detected multiple times over the course of a measurement (*Veatch et al., 2012*; *Stone et al., 2017*).

The cross-correlation analysis used here has advantages over other co-clustering algorithms for images with low spatial sampling, and can detect enrichment or depletion of probes even when this effect is not evident by visual inspection of images, as illustrated in Materials and methods. While in principle the cross-correlation analysis is insensitive to probe surface densities beyond an impact on signal to noise (*Figure 1—figure supplement 3*) (*Veatch et al., 2012*), we observe a weak expression level dependence of PM recruitment to CTxB clusters, possibly suggesting that probe expression impacts the mixing properties of the plasma membrane as a whole (*Figure 1—figure supplement 6*). Both the expression-level dependence of probe partitioning and the small size of CTxB and TM clusters impede us from drawing quantitative conclusions regarding the magnitude of enrichment or depletion of peptides into domains. Instead, we draw conclusions regarding whether

probes are enriched or depleted within our sensitivity limits, which is not impacted by either of these factors.

Taken together, these findings indicate that clustered plasma membrane proteins can stabilize domains spanning both plasma membrane leaflets that sort established markers of ordered and disordered domains in intact cell membranes. Importantly, we observe depletion of markers from domains of the alternate phase, as well as equivalency between order-and disorder-driven sorting. These are properties of liquid-ordered and liquid-disordered domains in phase separated membranes (*Veatch and Keller, 2005*); therefore we refer to them as phase-like domains.

## Phase-like domains are stabilized through BCR clustering

We used similar methods to probe membrane heterogeneity in the vicinity of BCR and BCR receptor clusters. Endogenously expressed BCR was labeled with a biotinylated f(Ab)$_1$ against IgM, BCR was clustered with streptavidin acting as a generic antigen, and BCR clusters were imaged in combination with transiently expressed PM or TM peptides (*Figure 2*). The sorting of phase sensitive peptides with respect to BCR clusters was observed in chemically fixed CH27 B cells (*Figure 2a*), live CH27 B cells (*Figure 2b*), and chemically fixed primary mouse B cells (*Figure 2c*). In all cases we found that the PM peptide was enriched and the TM peptide was excluded from BCR clusters. Cross-correlation curves were aggregated from multiple single-cell measurements, and error bars indicate the SEM between curves generated from single cells (*Figure 2—figure supplement 1*). Variance in these measurements is dominated by probe sampling statistics (*Figure 2—figure supplement 2*) and lipid probe expression density only weakly impacts the cross-correlation between BCR and lipid probes (*Figure 2—figure supplement 3*). Again, we found that the expression level of phase sensitive peptides impacts the magnitude but not the sign of peptide partitioning with respect to clustered BCR, allowing us to determine the type of domain stabilized by BCR clusters if not its quantitative composition. The direct measurements of peptide sorting shown here are generally consistent with past FRET and biochemical isolation measurements that argued that clustered BCR resides within ordered membrane domains (*Cheng et al., 1999*; *Pierce, 2002*; *Sohn et al., 2006*, *2008b*). The association between clustered BCR and the ordered domain marker PM appears more sustained in our imaging measurements than was observed in past reports using FRET (*Sohn et al., 2006*, *2008b*), possibly due to the different length-scales probed by these methods.

We find that both PM enrichment and TM depletion extend beyond BCR clusters themselves (*Figure 2—figure supplement 4*). It is likely that this extended domain arises from additional signaling structures assembled proximal to BCR, as is observed in other immune-receptor signaling systems (*Balagopalan et al., 2015*). For example, palmitoylated adapter proteins involved in signal transduction such as LAB/LAT2/NTAL may incorporate into activated BCR microclusters and act to extend the domain (*Mutch et al., 2007*; *Malhotra et al., 2009*). Further, PM enrichment was reduced in cells treated with the Src kinase inhibitor PP2 prior to receptor clustering and fixation (*Figure 2—figure supplement 5*), suggesting that ordered domain stabilization is amplified by receptor activation and the recruitment of down-stream signaling partners.

The same magnitude of PM and TM co-localization with BCR clusters was observed in live cells (*Figure 2b*) as in chemically fixed cells, indicating that co-localization is not an artifact of chemical fixation. Here co-localization was quantified using a steady-state cross-correlation function (*Stone and Veatch, 2015*). Additional sensitivity was obtained in these measurements by including probe pairs imaged within a time separation of up to 50 frames or approximately 1 s since we did not observe significant changes in steady state correlations over this window (*Figure 2—figure supplement 6*). We note that PM proximal to BCR did not exhibit altered mobility in live cells, indicating that PM enrichment arises from weak and/or transient interactions (*Figure 2—figure supplement 7*). As a counter-example, Lyn proximal to BCR does exhibit slowed diffusion, likely due to specific Lyn-BCR binding interactions. *Videos 1–3* show single molecule localizations compiled over time for BCR-PM, BCR-TM, and BCR-Lyn, respectively. These videos demonstrate that the distributions of PM and TM do not change dramatically upon BCR clustering.

We also observed sorting of PM and TM peptides with respect to BCR clusters imaged in primary mouse B cells fixed 5 min following antigen stimulation (*Figure 2c*) that is qualitatively consistent with observations in the CH27 cell line. Interestingly, the magnitude of sorting is increased in primary cells compared to CH27 cells. This may be a biological consequence of the LPS treatment required to maintain cell viability during transient transfection, or due to other differences in membrane

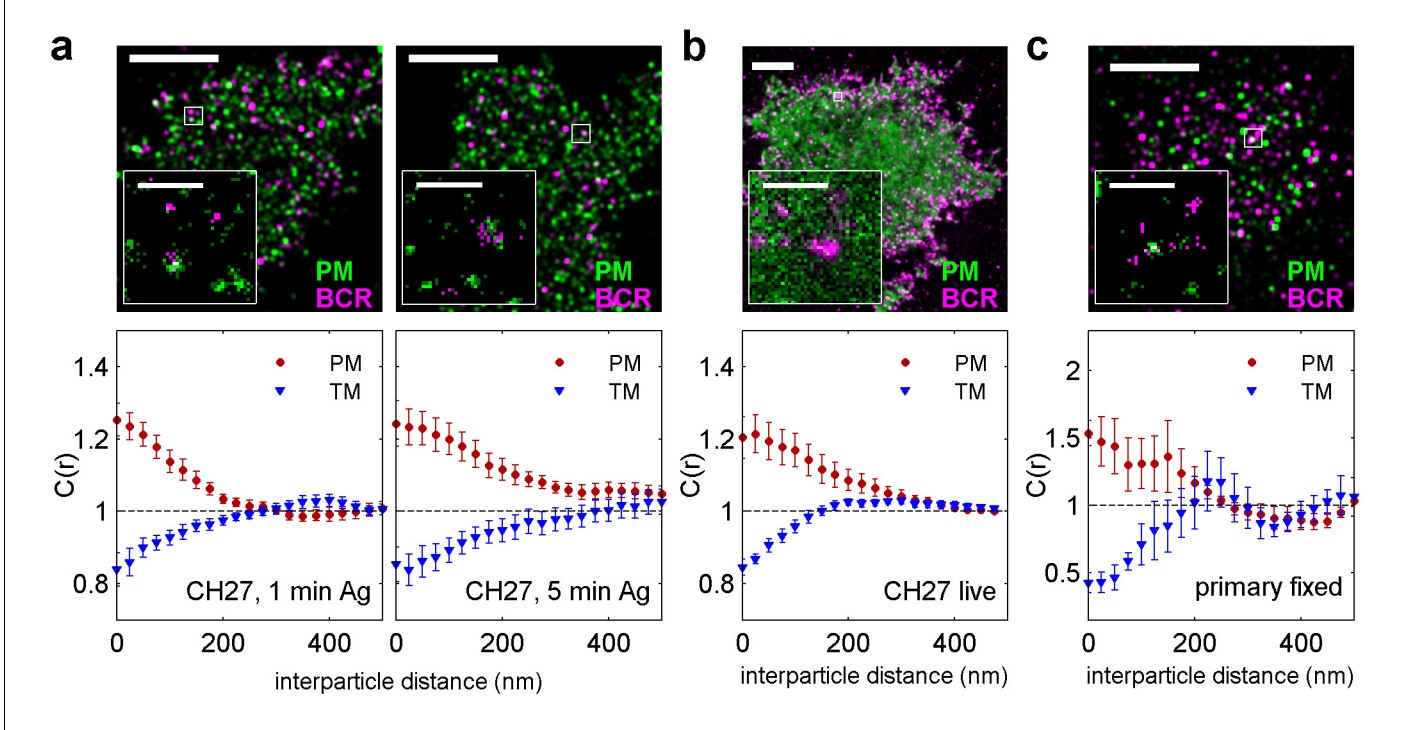

**Figure 2.** BCR clusters localize within ordered membrane domains. (Upper panels) Representative reconstructed super-resolution images of the BCR and PM in chemically fixed (a) and live (b) CH27 B cells, and chemically fixed primary B cells (c). Scale-bars are 5 μm and 500 nm in the inset. (Lower panels) Average cross-correlation curves, C(r), between BCR and phase markers. Error-bars indicate the SEM between cells. In (a) cells were chemically fixed either 1 min following BCR clustering (1 min Ag, left) or 5 min after BCR clustering (5 min Ag, right). In (b), data was acquired from live cells between 0 and 6 min following BCR clustering. In (c), BCR was clustered for 5 min prior to chemical fixation. In all cases, the order-favoring peptide (PM) was enriched and the disorder-favoring peptide (TM) was depleted from BCR clusters. Curves from individual cells are shown in *Figure 2—figure supplement 1*. Curves are averaged over the following number of cells: (a) 1 min BCR and PM (18) or TM (11); 5 min BCR and PM (21) or TM (10). (b) BCR and PM (4) or TM (4). (c) BCR and PM (4) or TM (5). Correlation curves from right column in (a) were used to make a schematic figure showing enrichment and depletion of probes around BCR clusters in *Figure 2—figure supplement 4*. Representative images for conditions not shown here can be found in *Figure 2—figure supplement 10*.

The following figure supplements are available for figure 2:

**Figure supplement 1.** Correlation functions from individual cells and average curves.

**Figure supplement 2.** Distribution of correlation function values closely matches the width expected from single measurement errors.

**Figure supplement 3.** Dependence of cross-correlation amplitudes on lipid probe expression levels.

**Figure supplement 4.** PM and TM cross-correlation functions have a larger correlation length than BCR autocorrelation functions.

**Figure supplement 5.** Cross-correlations between clustered BCR and PM are reduced in the presence of a signaling inhibitor.

**Figure supplement 6.** Cross-correlations in live cells are calculated by averaging correlations between non-simultaneous frames.

**Figure supplement 7.** The mobility of lipid probes is not altered when in close proximity to BCR clusters.

**Figure supplement 8.** Cross-correlations between clustered BCR and PM are reduced but still observable at physiological temperatures.

**Figure supplement 9.** Cross-correlations between PM and unclustered BCR or CTxB are near detection limits.

**Figure supplement 10.** Representative images from *Figure 2*.

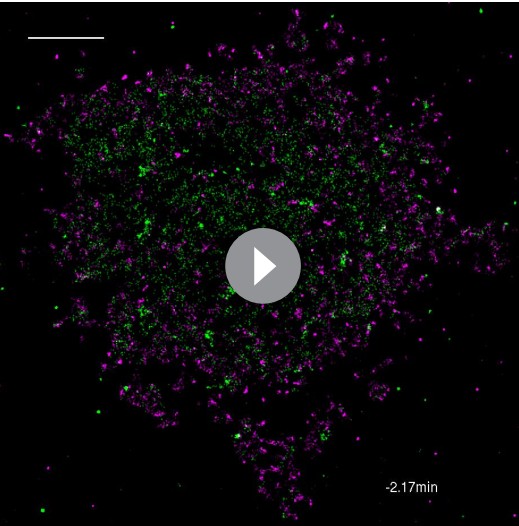

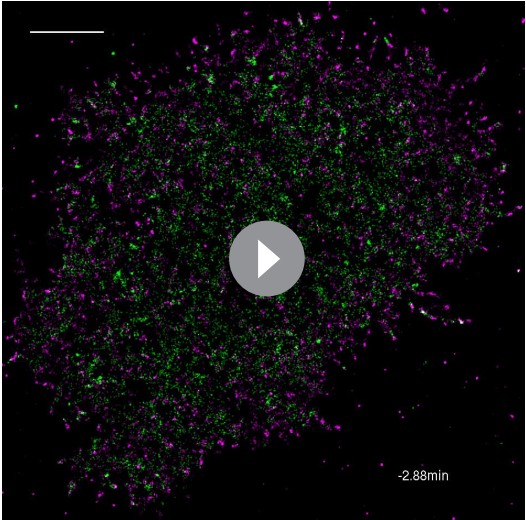

**Video 1.** Reconstructed image time lapse of single molecule localizations from live cell measurements of BCR (magenta) and PM (green). Individual images are reconstructed using 100 frames (2 s) of single molecule images, receptors are crosslinked at time = 0, and the scale-bar is 5 μm. Single BCR or PM proteins are imaged over multiple frames and each produce clouds of localizations whose extent depend on the protein mobility as well as the typical time that a probe remains activated. This combined with under-sampling of the molecules that are present gives rise to the self-clustered appearance of probes (especially BCR) prior to receptor clustering. Overall, there is not an obvious reorganization of PM after BCR is clustered.

**Video 2.** Reconstructed image time lapse of single molecule localizations from live cell measurements of BCR (magenta) and TM (green). Individual images are reconstructed using 100 frames (2 s) of single molecule images, receptors are crosslinked at time = 0, and the scale-bar is 5 μm. Single BCR or TM proteins are imaged over multiple frames and produce clouds of localizations whose extent depend on the protein mobility as well as the typical time that a probe remains activated. This combined with under-sampling of the molecules that are present gives rise to the self-clustered appearance of probes (especially BCR) prior to receptor clustering. Overall, there is not an obvious reorganization of PM after BCR is clustered.

composition between these two cell types. PM was also enriched within BCR clusters when CH27 cells were chemically fixed at 37°C (*Figure 2—figure supplement 8*), indicating that ordered domains are formed at growth temperatures, although the magnitude of enrichment is reduced compared to the room temperature examples shown in *Figure 2a*. We note that BCR clustering elicits a cellular response at both temperatures.

We additionally observed weak PM enrichment around unclustered BCR in chemically fixed CH27 cells, although the magnitude of this sorting is on the edge of the sensitivity limits of our imaging system (*Figure 2—figure supplement 9*). This weak signal could indicate reduced partitioning of monomeric or pre-clustered BCR with ordered domains prior to streptavidin-induced clustering, or could simply reflect that domain sizes are at or below the finite lateral resolution of these measurements (~40 nm) (illustrated in *Figure 1—figure supplement 3*). At this time, we cannot comment on the oligomerization state of BCR prior to enforced clustering with streptavidin, because single color images of BCR contain over-counting artifacts that render monomers indistinguishable from small oligomers (*Veatch et al., 2012*). Interestingly, the magnitude of PM sorting with respect to unclustered BCR is comparable to the sorting of this peptide observed with respect to unclustered CTxB (*Figure 2—figure supplement 9*). Improved lateral resolution is needed to systematically investigate the lateral organization of receptors and peptides in intact cells without receptor clustering, and may be enabled by recent improvements in fluorophores and imaging modalities (*Huang et al., 2013*; *Grimm et al., 2015*, *2016*).

Together, the results presented in *Figures 1* and *2* support the view that BCR clustering locally stabilizes an ordered lipid domain. Sorting of order- and disorder-preferring probes around BCR

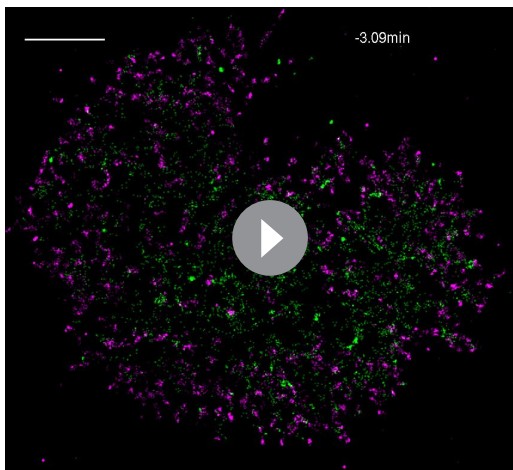

**Video 3.** Reconstructed image time lapse of single molecule localizations from live cell measurements of BCR (magenta) and Lyn kinase (green). Individual images are reconstructed using 100 frames (2 s) of single molecule images, receptors are crosslinked at time = 0, and the scale-bar is 5 μm. Single BCR or Lyn proteins are imaged over multiple frames and produce clouds of localizations whose extent depend on the protein mobility as well as the typical time that a probe remains activated. This combined with under-sampling of the molecules that are present gives rise to the self-clustered appearance of probes (especially BCR) prior to receptor clustering. A population of Lyn is immobile or diffuses more slowly. These proteins appear as brighter spots since numerous localizations occur in the same location.

clusters follows the same trends observed for clusters of an established marker of ordered domains, CTxB, and is consistent in fixed, live, and primary cells. Domains formed around clustered proteins share many of the features of domains observed in model membranes. Individual lipids are free to diffuse into and out of domains, suggesting that these domains are fluid. The amount of enrichment observed in these experiments is also consistent with expectations from ordered-disordered phase fluctuations in model systems above their transition temperatures (*Veatch et al., 2008*; *Machta et al., 2012*; *Zhao et al., 2013*), in contrast to the strong recruitment described in early work on membrane rafts (*Simons and Ikonen, 1997*). Finally, the symmetry in the sign of correlations observed for order- and disorder-preferring probes and order- or disorder-preferring protein clusters is also consistent with the two liquid phase regime in model systems (*Machta et al., 2012*). At the same time, the effects of lipid phase-mediated sorting are likely superimposed on other interactions present within the complex milieu of the membrane, such as electrostatic interactions between charged peptides and lipids.

## Ordered domains sort key regulators of early BCR signaling

The observed sorting of minimal peptides suggests that the localization of full-length proteins is also impacted by their membrane anchoring motifs. We investigated the role of membrane domains in the localization of two critical regulators of BCR activation, Lyn kinase and CD45 phosphatase (*Dal Porto et al., 2004*), by comparing their spatial distribution around BCR to that of their membrane anchor peptides (*Figure 3a*). Both CD45 and its minimal anchor peptide CD45$_{TM}$ were excluded from BCR clusters to the same extent, indicating that membrane domain interactions are sufficient to drive this partitioning. Lyn kinase was more enriched in BCR clusters than its membrane anchor PM, consistent with the direct binding of Lyn to phosphorylated ITAM sequences on BCR mediated by SH2 interaction domains in full-length Lyn (*Johnson et al., 1995*). The effective interaction energy between membrane anchors and BCR is given by the potential of mean force (PMF), which is simply obtained from measured cross-correlation functions (*Veatch et al., 2012*; *Stone and Veatch, 2015*). PMFs between the BCR and PM or Lyn indicate that the anchor sequence provides over one third of the energy associated with Lyn-BCR interactions (*Figure 3b*).

We demonstrated that ordered domains are sufficient to sort full length signaling proteins by also monitoring their distribution with respect to CTxB clusters (*Figure 3a*). Again we found that CD45 was depleted and Lyn was enriched in these ordered domains. In this case, full-length proteins partitioned similarly to their minimal anchor peptides, as expected since CTxB does not bind directly to either of these proteins.

The sorting of proteins into ordered membrane domains is also sufficient to locally alter their phosphorylation state. This was seen by visualizing the distribution of phosphorylated protein tyrosine residues (pY) with respect to protein clusters using a generic anti-pY antibody (*Figure 3c*). BCR and CTxB clustering both induce local tyrosine phosphorylation co-localized with clusters, while TM clustering does not. This direct observation agrees with previous indirect measures of localized phosphorylation (*Harder and Simons, 1999*; *Cheng et al., 1999*). As expected, total pY enrichment was

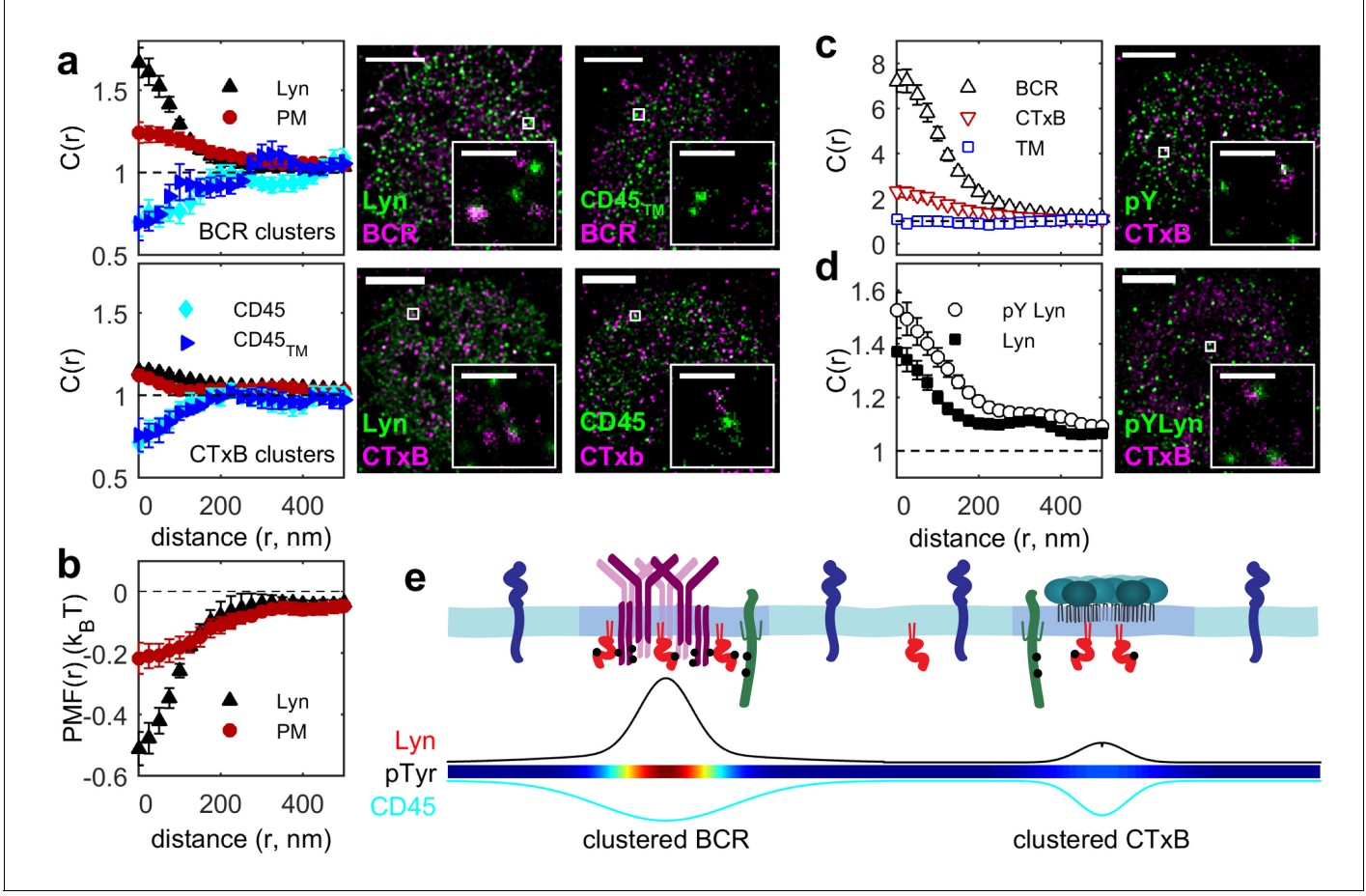

**Figure 3.** Ordered domains promote tyrosine phosphorylation. (**a**) Average cross-correlation functions (C(r), left) and representative super-resolution images (right) demonstrating that full-length proteins and their minimal membrane anchors sort with respect to clusters of both BCR (top) and CTxB (bottom). The correlations presented represent an average over multiple (N) individual cells: Between BCR and Lyn (4), PM (21), CD45 (10), CD45$_{TM}$ (10); between CTxB and Lyn (19), PM (58), CD45 (11), and CD45$_{TM}$ (5). (**b**) Lyn and PM distributions with respect to B cell receptor clusters expressed as the potential of mean force (PMF). (**c**) Both BCR and CTxB domains are sites of tyrosine phosphorylation (pY) as detected through a generic anti-pY antibody (4G10), while disordered TM domains were not enriched in pY proteins. Curves are averaged over the following number of cells: BCR and pY (14), CTxB and pY (13), and TM and pY (7). (**d**) Activated Lyn (pY-397) was more enriched in CTxB clusters than Lyn as a whole, indicating that ordered domains favor activation of this protein. Curves are averaged over the following number of cells: CTxB and pY Lyn (40), CTxB and Lyn (49). (**e**) Schematic of membrane domains stabilized by BCR and CTxB clusters. Curves and color-scale at bottom are quantitative representations of the relative enrichment or depletion of the components indicated as represented in parts **a** and **c**. Scale-bars are 5 μm and 500 nm in the inset. Additional representative images are shown in *Figure 3—figure supplement 4*.

The following figure supplements are available for figure 3:

**Figure supplement 1.** Cell surface clustering of cholera toxin subunit B elicits calcium mobilization in B cells.

**Figure supplement 2.** CTxB clusters are not highly correlated with BCR.

**Figure supplement 3.** Subtle increases in protein phosphotyrosine levels in response to CTxB clustering are suggested by western blots of whole cell lysates.

**Figure supplement 4.** Representative images from *Figure 3*.

reduced in CTxB clusters compared to BCR clusters, since CTxB itself lacks sites for tyrosine phosphorylation and does not directly bind to additional proteins containing pY. However, enrichment in ordered domains stabilized by CTxB clustering is sufficient for phosphorylation of resident proteins,

as is evident from the stronger co-localization of trans-activated (pY397) Lyn with CTxB clusters compared to overall Lyn (*Figure 3d*). The membrane remodeling that occurs upon CTxB clustering is also sufficient to trigger a cellular response. In agreement with past reports (*Francis et al., 1992*), we found that CTxB clustering leads to $Ca^{2+}$ mobilization (*Figure 3—figure supplement 1*) without significantly altering the distribution of BCR (*Figure 3—figure supplement 2*). CTxB binding and clustering also resulted in subtle increases in tyrosine phosphorylation of multiple protein species detected within cellular extracts probed via Western blot (*Figure 3—figure supplement 3*).

The imaging results shown in *Figure 3* draw a connection between the lipid-mediated protein sorting observed in *Figures 1* and *2* and signaling function. We showed that ordered domains contribute a substantial fraction of the free energy required to concentrate the kinase Lyn and all of the free energy required to deplete the phosphatase CD45 from BCR clusters. Through this sorting, ordered domains provide a local environment that favors tyrosine phosphorylation, as supported by correlations between CTxB clusters and anti-pY antibodies even in the absence of specific recruitment of signaling machinery through protein-protein interactions. It is reasonable to expect that the ~50% increase in the ratio of Lyn to CD45 due to sorting by ordered domains could cause a significant change in BCR phosphorylation levels if we compare to results obtained for the related T cell receptor (TCR) system. In a reconstituted system, TCR phosphorylation was shown to have switch-like dependence on the relative concentrations of the Src kinase Lck, which is the analog of Lyn in the TCR system, and CD45 at physiological levels (*Hui and Vale, 2014*). As a result, small increases in Lck concentration and decreases in CD45 concentration were shown to produce large shifts in TCR phosphorylation, and this behavior likely also applies to the BCR system investigated here. Additionally, the actual enrichment and depletion of probes around BCR and CTxB is larger than the measured values presented here since the real spatial distributions are convolved with the finite resolution of the measurement to give the observed cross-correlations. Lastly, local activation of Lyn within ordered domains also provides a potential mode of positive regulation. Concentration of Lyn in ordered domains and exclusion of phosphatases would be expected to favor Lyn trans-activation at Y397 and prevent inactivation. This would create an environment within ordered domains where Lyn is not only more concentrated but also more active, as has been suggested previously (*Young et al., 2003*) and supported by the results shown in *Figure 3d*. Thus, distinct membrane environments could influence both the local concentration and activity of proteins, and these effects may amplify or negate one another to determine an overall signaling outcome.

## A minimal model for receptor activation upon clustering in a heterogeneous membrane

Our observations of protein sorting by ordered domains suggest a mechanism for receptors to become phosphorylated upon clustering via differential partitioning of proteins regulating BCR phosphorylation. *Figure 4* describes a predictive model that reproduced the sorting behavior of kinase and phosphatase anchor peptides observed experimentally (*Figure 4a* and *Figure 4—figure supplement 1*). The model consists of receptors that can be phosphorylated by kinases and dephosphorylated by phosphatases. In addition, activated receptors can phosphorylate other receptors, mimicking the actions of receptor-bound kinases (RBKs) such as Lyn and Syk (*Johnson et al., 1995*). These protein components are embedded in a heterogeneous membrane represented by a 2D Ising model where extended ordered and disordered domains form at equilibrium through interactions between adjacent components (*Machta et al., 2011*, *2012*). This model represents phase-like heterogeneity as extended composition fluctuations that collectively emerge from weak intermolecular interactions when membranes are positioned near a miscibility critical point, and is supported by experiments in both purified and isolated biological membranes (*Veatch et al., 2008*; *Honerkamp-Smith et al., 2008*; *Zhao et al., 2013*). Receptors and kinases act as typical ordered components and phosphatases act as typical disordered components. Through this set of minimal assumptions, receptors became collectively activated upon clustering (*Figure 4b* and *Video 4*). Clustered receptors were activated to a lesser extent in the absence of RBK positive feedback, but were not activated in a uniform membrane even with RBK feedback (*Figure 4b* and *Videos 5* and *6*). This localized receptor phosphorylation may favor recruitment and assembly of adapter proteins that mediate the cellular immune response (*Su et al., 2016b*).

Receptor activation was also observed when an ordered domain was stabilized using an external potential without confining receptors to the domain (*Figure 4c*), mimicking the CTxB clustering

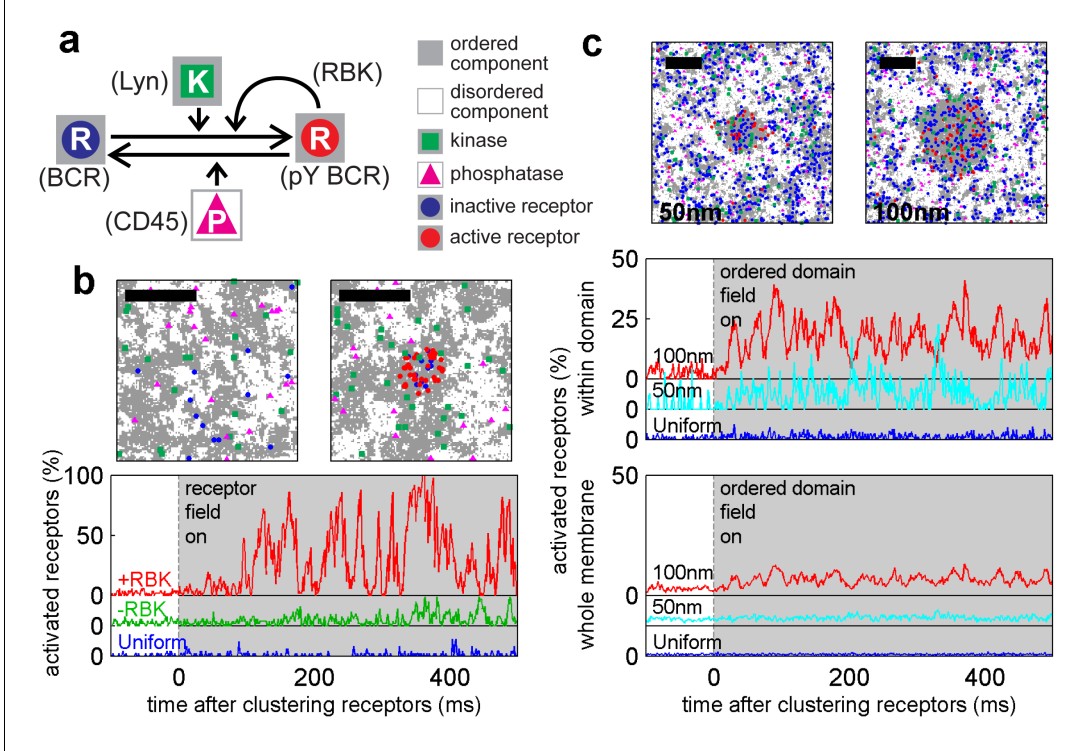

**Figure 4.** A model linking receptor clustering to receptor phosphorylation. (**a**) Schematic representation of the model described in the main text. Possible biological analogs of model components are indicated, with "RBK" representing receptor-bound kinases. (**b**) Simulation snap-shots (top) and sample receptor phosphorylation time-traces (bottom) indicate receptors are robustly phosphorylated (pY) upon clustering when membranes are heterogeneous. Phosphorylation was diminished in simulations without RBKs and absent in simulations with a uniform membrane. All curves have the same vertical scale and time-lapses of the simulations are in *Videos 4–6*. (**c**) Simulation snap-shots (top) and sample receptor phosphorylation time-traces (bottom) for simulations where an ordered domain is stabilized without confining receptors to the domain. In this case, more receptors are included to represent the large number of membrane proteins containing tyrosine residues that can be phosphorylated. The efficiency of receptor phosphorylation depends on the size of the stabilized ordered domain for reasons discussed in the main text. All curves have the same vertical scale. Time-lapses of the simulations are shown in *Videos 7–9*. All scale bars are 100 nm.

The following figure supplement is available for figure 4:

**Figure supplement 1.** Simulations naturally reproduce experimental kinase and phosphatase distributions with respect to the BCR.

result of *Figure 3*. Receptor activation was more subtle in these simulations. This is because only a fraction of receptors reside within the ordered domain at a given time, receptors freely leave the protective environment of the ordered domain, and the local concentration of receptors within the domain is reduced compared to simulations with clustered receptors. In these simulations, we also find that the size of the ordered domain impacts the efficiency of receptor activation. This occurs because the residency time of a receptor within the domain increases with increasing domain size, as does the average number of receptors within the domain. Both of these factors increase the likelihood that the RBK positive feedback will amplify receptor activation.

This minimal model provides a framework for interpreting our imaging data and for developing hypotheses for the specific role of lipid-mediated sorting in BCR clustering-induced activation. Here, existing small, dynamic phase fluctuations were stabilized through clustering of ordered components to form a larger, stable domain structure of approximately the size of the cluster. This model quantitatively reproduces our imaging measurements of protein sorting by ordered domains, recapitulates clustering-induced BCR activation, and demonstrates that ordered domain formation alone can be sufficient to activate receptors. Further, phase-like heterogeneity is required for robust activation of BCR under these simulation conditions. The dependence of receptor activation on domain size in

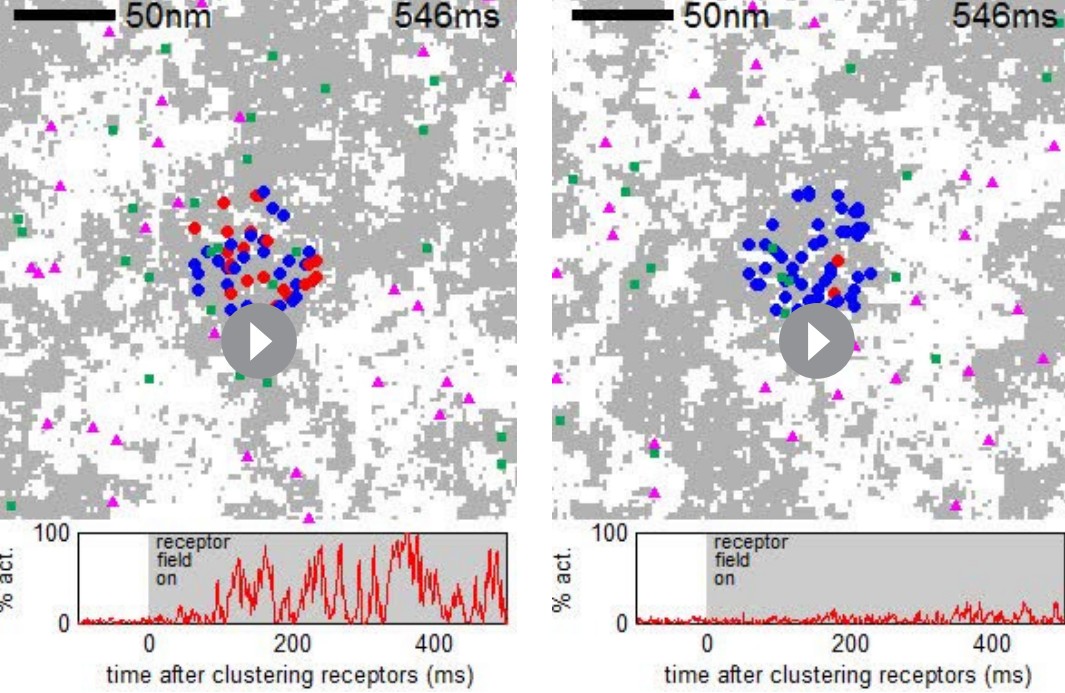

**Video 4.** Simulated time course of receptor activation upon clustering in a heterogeneous membrane. Simulations are conducted as described in Methods. The positions of receptors (circles), kinases (green squares), and phosphatases (magenta triangles) are shown at 1 ms intervals (representing 1000 simulation updates). Inactive receptors are shown in blue and activated (phosphorylated) receptors are shown in red. Symbols are drawn larger than the pixels that they represent for clarity. The receptor field is turned on at time = 0 to induce receptor clustering. The time trace shown at the bottom is redrawn from *Figure 4b*.

**Video 5.** Simulated time course of receptor activation upon clustering in a heterogeneous membrane without the positive feedback loop accomplished through receptor bound kinases. Simulations are conducted as described in Methods. The positions of receptors (circles), kinases (green squares), and phosphatases (magenta triangles) are shown at 1 ms intervals (representing 1000 simulation updates). Inactive receptors are shown in blue and activated (phosphorylated) receptors are shown in red. Symbols are drawn larger than the pixels that they represent for clarity. The receptor field is turned on at time = 0 to induce receptor clustering. The time trace shown at the bottom is redrawn from *Figure 4b*.

simulations where ordered domains are formed without BCR clustering (*Figure 4c*) highlights the collective nature of interactions that determine the local environment of receptors.

## The minimal model is predictive for the case of cholesterol modulation

In order to demonstrate this model's predictive power, we ran further simulations to probe receptor phosphorylation as the surface fraction of ordered components was varied (*Figure 5a*). Clustered receptors were more phosphorylated in simulations with a larger fraction of disordered components and were less phosphorylated in simulations with more ordered components. This occurs because varying the surface fraction of ordered vs. disordered components acts to enhance (or suppress) the local enrichment and depletion of signaling modulators at receptor clusters (*Figure 5—figure supplement 1*). This is also reflected in cross-correlation functions calculated from simulations (*Figure 5b*) that report the co-localization of receptors and the order-preferring kinase for the surface fractions indicated.

These simulation results are qualitatively consistent with functional and imaging data obtained in B cells with modulated cholesterol levels (*Figure 5c,d* and *Figure 5—figure supplement 2*). Acute modulation of cholesterol levels in intact cells with methyl $\beta$ cyclodextrin (M$\beta$CD) alters the surface

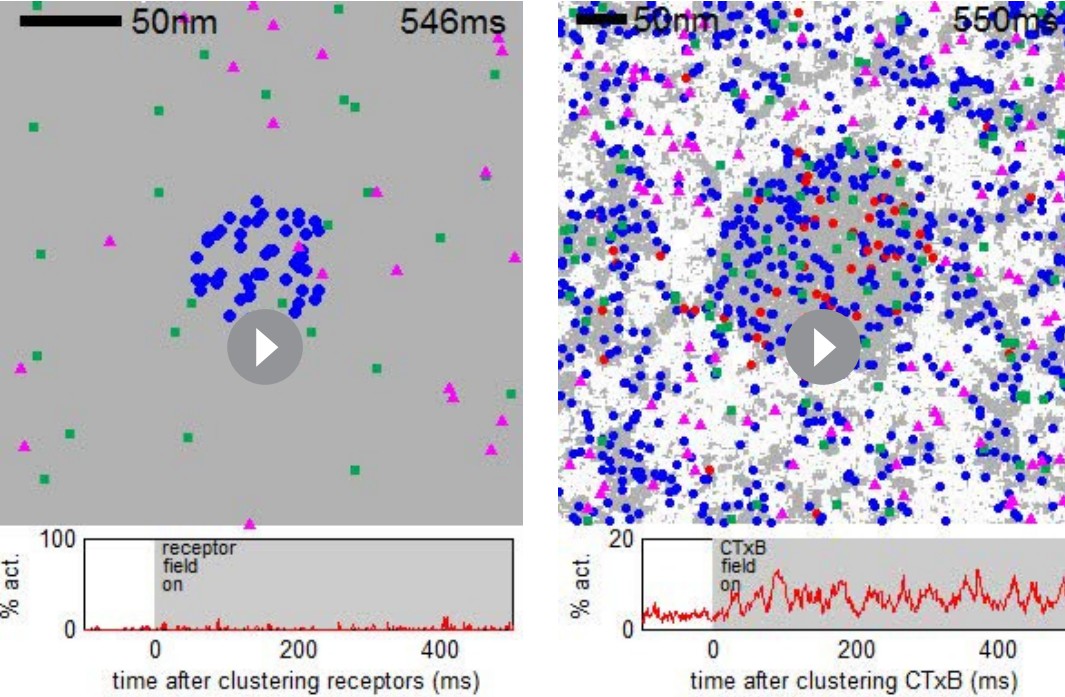

**Video 6.** Simulated time course of receptor activation upon clustering in a uniform membrane. Simulations are conducted as described in Methods. The positions of receptors (circles), kinases (green squares), and phosphatases (magenta triangles) are shown at 1 ms intervals (representing 1000 simulation updates). Inactive receptors are shown in blue and activated (phosphorylated) receptors are shown in red. Symbols are drawn larger than the pixels that they represent for clarity. The receptor field is turned on at time = 0 to induce receptor clustering. The time trace shown at the bottom is redrawn from **Figure 4b**.

**Video 7.** Simulated time course of receptor activation upon stabilization of a large ordered domain. Simulations are conducted as described in Methods. The positions of receptors (circles), kinases (green squares), and phosphatases (magenta triangles) are shown at 1 ms intervals (representing 1000 simulation updates). Inactive receptors are shown in blue and activated (phosphorylated) receptors are shown in red. Symbols are drawn larger than the pixels that they represent for clarity. The field is turned on at time = 0 to induce a circular ordered domain with a radius of 100 nm. The time trace shown at the bottom is redrawn from **Figure 4c**.

fraction of coexisting ordered and disordered phases in GPMVs imaged at low temperature that were isolated from these cells (**Figure 5c**). This occurs without an effect on cell viability as measured via annexin V binding to the extracellular surface (**Figure 5—figure supplement 3**). Past work indicates that this perturbation can also alter domain composition in vesicles at elevated temperature in a way that is quantitatively predicted by our model of membrane heterogeneity (**Zhao et al., 2013**). In intact B cells, we observed increased calcium mobilization upon BCR clustering when cholesterol levels were acutely lowered with MβCD, in agreement with past work (**Awasthi-Kalia et al., 2001**). We also observed decreased calcium mobilization upon BCR clustering in cells pretreated with MβCD-cholesterol complexes, which act to increase cellular cholesterol levels (**Christian et al., 1997**). The observed changes in calcium mobilization are accompanied by changes in PM cross-correlation with BCR clusters that are also in line with model predictions (**Figure 5d**). PM is more strongly enriched in BCR clusters in cells pretreated with MβCD than in untreated cells, and less strongly enriched in cells pretreated with MβCD-cholesterol complexes.

While the model clearly over-simplifies the behavior of the plasma membrane and the complexity of the BCR signaling pathway, it is able to predict both the functional response associated with membrane perturbation and changes in domain-mediated sorting reported by super-resolution imaging experiments. In spite of the acknowledged plieotropic effects of cholesterol modulation

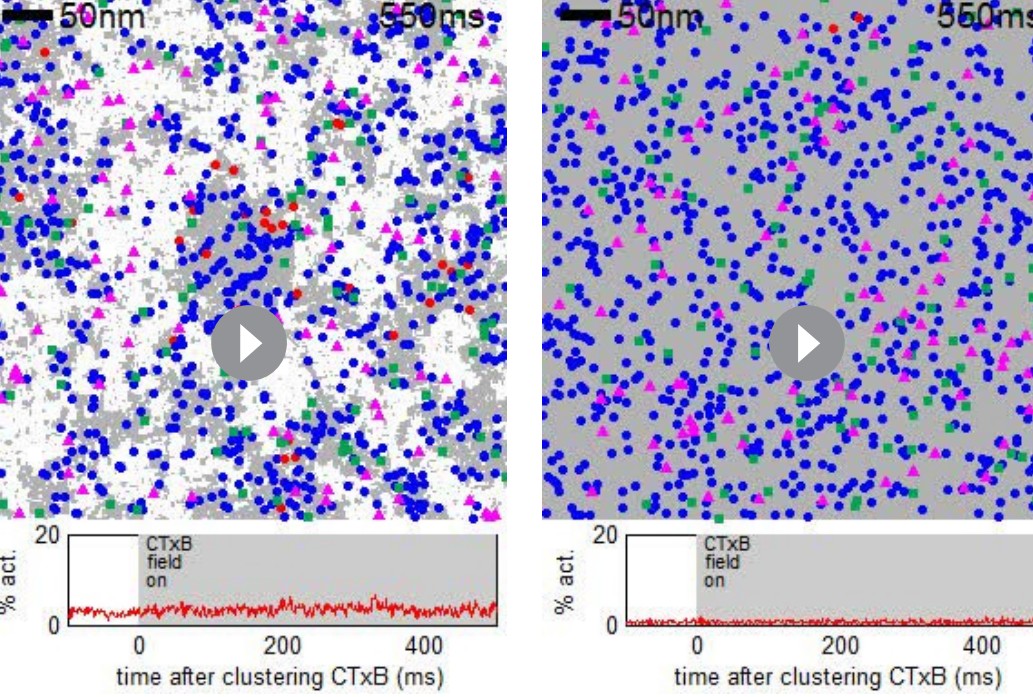

**Video 8.** Simulated time course of receptor activation upon stabilization of a small, ordered domain. Simulations are conducted as described in Methods. The positions of receptors (circles), kinases (green squares), and phosphatases (magenta triangles) are shown at 1 ms intervals (representing 1000 simulation updates). Inactive receptors are shown in blue and activated (phosphorylated) receptors are shown in red. Symbols are drawn larger than the pixels that they represent for clarity. The field is turned on at time = 0 to induce a circular ordered domain with a radius of 50 nm. The time trace shown at the bottom is redrawn from *Figure 4c*.

**Video 9.** Simulated time course receptor activation state within a uniform membrane with an applied field. Simulations are exactly as described for *Video 7* but unspecified membrane components are all of the same type (ordered in this case). The positions of receptors (circles), kinases (green squares), and phosphatases (magenta triangles) are shown at 1 ms intervals (representing 1000 simulation updates). Inactive receptors are shown in blue and activated (phosphorylated) receptors are shown in red. Symbols are drawn larger than the pixels that they represent for clarity. The field is turned on at time = 0 but does not impact the organization of components. The time trace shown at the bottom is redrawn from *Figure 4c*.

with M$\beta$CD (*Munro, 2003*; *Kwik et al., 2003*), its effects on the surface fraction of ordered vs. disordered phases in GPMVs and the cross-correlation of PM with BCR clusters mirror the changes in the fraction of ordered and disordered components enforced in simulations. Therefore, we expect that modulating cholesterol levels using M$\beta$CD impacts the composition of the plasma membrane of B cells in a way that parallels the changes in our minimal model: by varying the surface fraction of ordered vs. disordered components. This in turn alters peptide sorting around BCR clusters. The striking correspondence of the structural and functional outputs of the simulations to experimental measures of PM partitioning and BCR activation provides validation of the model and its assumptions.

## Conclusions

In conclusion, we directly observed domains in cellular plasma membranes that resemble liquid-ordered and liquid-disordered phases in model membranes. In these experiments we visualized both enrichment and, for the first time, depletion of membrane species from phase-like domains in intact cells. This sorting behavior was induced both by clustering generic membrane proteins with known phase partitioning in GPMVs and by clustering BCR. BCR clustering drove spatial co-

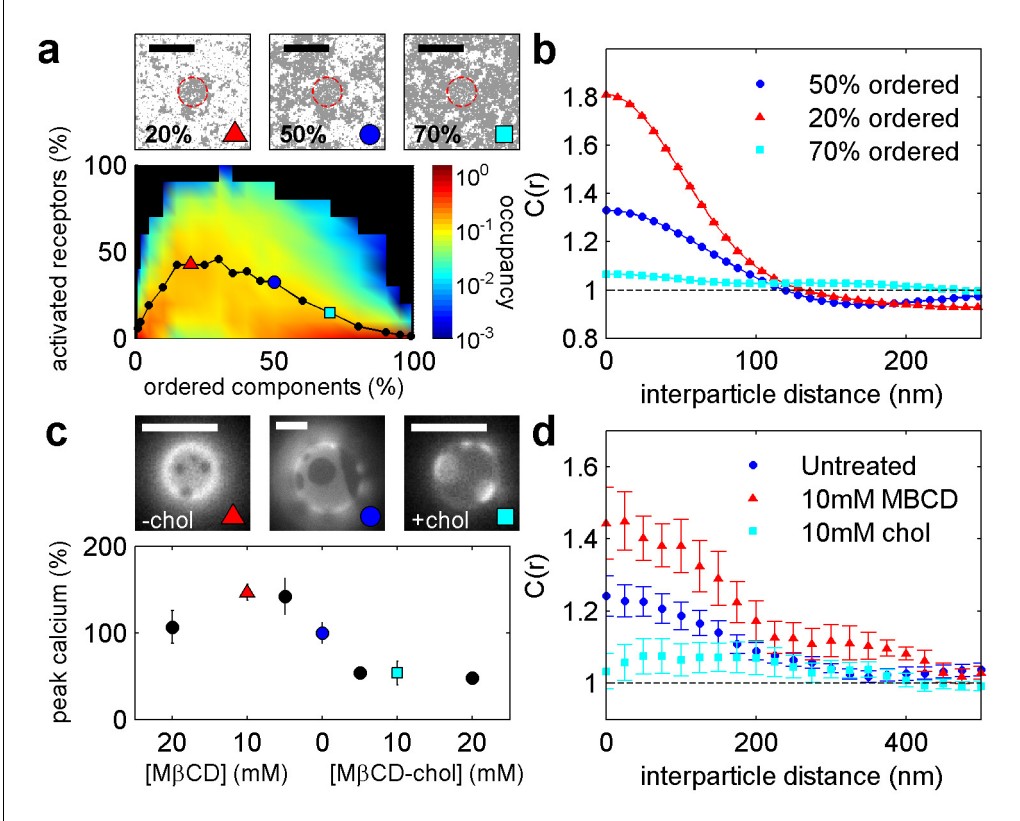

**Figure 5.** Phosphorylation model predicts the response to changing the fraction of ordered and disordered components. (a) Representative snap-shots (top) and histograms showing receptor phosphorylation (bottom) in simulations run with different fractions of ordered and disordered components (grey and white pixels respectively). Receptors (not shown) were confined within the red dashed circle. Snap-shots corresponding to conditions in the plot are indicated as colored symbols. (b) Cross-correlations between receptors and ordered components in simulations run with a variable fraction of ordered and disordered components. Simulations shown are a subset of those displayed in (a) as indicated by symbols. (c) Acute cholesterol modulation with MβCD altered the surface fraction of ordered (dark) and disordered (bright) phases in isolated GPMVs (top) and modulated calcium mobilization in response to antigen, measured using the cytoplasmic Ca²⁺ indicator Fluo-4 (bottom). Values were normalized to the maximum response of untreated cells. Colored symbols on vesicle micrographs and plot indicate equivalent treatments in both measurements. Calcium measurements are an average of at least two biological replicates. (d) Cross correlations between BCR and PM-mEos3.2 in CH27 cells treated with indicated amounts of MβCD (18), cholesterol-loaded MβCD (18), or left untreated (21) and then fixed 5 min following antigen addition. Scale bars are 100 nm in simulations and 5 µm in micrographs.

The following figure supplements are available for figure 5:

**Figure supplement 1.** Kinase and phosphatase partitioning into receptor clusters change dramatically as the surface fraction of ordered components is varied.

**Figure supplement 2.** Averaged and baseline-corrected Fluo-4 intensity curves.

**Figure supplement 3.** Cholesterol treatments do not alter annexin V staining.

localization or segregation of minimal membrane anchors from BCR clusters, consistent with the formation of an ordered phase-like domain. These domains also sorted BCR signaling partners into and out of receptor clusters. Experiments that examined functional outcomes of BCR signaling indicated that formation of an ordered domain through protein clustering creates a local membrane

environment that favors protein tyrosine phosphorylation. Functional and imaging results are consonant with a simple but predictive model of membrane phase behavior.

Based on these observations and supported by the predictive model, we propose a mechanism where receptor clustering alone can initiate collective BCR activation by stabilizing an ordered membrane domain. This mechanism is distinct from some historical ideas of how "lipid rafts" have been thought to play a role in BCR signaling, which pose that BCR actively partitions into pre-existing ordered domains upon clustering. Instead, our findings support the idea that receptor clusters template a lipid domain in the membrane by stabilizing existing but very subtle and transient membrane composition fluctuations. In the absence of clustering, the size and dynamics of fluctuations allows positive and negative regulators to mix with BCRs. When BCR is clustered, the effect on downstream signaling depends on the size and stability of the resulting domain, which sets the level of access signaling partners have to resident BCR. As a consequence, the structure of the BCR cluster and surrounding membrane can influence the activation state of individual BCRs. Thus, the entire cluster and associated domain, as opposed to single BCRs, comprise the basic signaling unit for the activation of B cells in this context. Super-resolution microscopy is particularly well-suited to capture this action-at-a-distance because it can measure interactions that span the length scale of the BCR cluster. While BCR activation in vivo by natural antigens likely involves signal integration from multiple sources, our work suggests that formation of phase-like ordered domains through a collective ensemble of relatively weak protein-lipid and lipid-lipid interactions impacts overall signaling outcomes. Thus, we both conclude that lipid-mediated interactions can play a significant role in signal transduction from BCRs activated through protein clustering and can describe a plausible mechanism for their action.

We suggest that the signaling function conferred by formation of phase-like domains around BCR clusters is an emergent property of a system where signaling molecules are compartmentalized based on their interactions with plasma membrane lipids. By extension, these findings suggest that membrane domains are capable of contributing to a broad array of signal transduction pathways by altering the local concentration of regulatory proteins, shifting the balance of biochemical networks. This compliments existing theories that invoke spatial compartmentalization of receptors and regulatory molecules as a means to control receptor signaling, including size-based exclusion (*Choudhuri et al., 2005*; *Cordoba et al., 2013*) and 3D protein phase separation (*Su et al., 2016a*) and places lipid-mediated interactions (*Machta et al., 2012*) among other canonical forces that dictate protein interactions in membranes, such as electrostatics, curvature, and adhesion.

## Materials and methods

### f(Ab)$_1$ and antibody modification

f(Ab)$_1$ fragment goat antibody to mouse IgM, μ chain specific (Jackson ImmunoResearch, West Grove, PA; RRID: AB_2338477) was simultaneously chemically modified with Atto 655 NHS ester (Sigma, St. Louis, MO) and biotin-X, SSE, 6-((Biotinoyl)Amino)Hexanoic Acid, Sulfosuccinimidyl Ester, Sodium Salt (Sulfo-NHS-LC-Biotin) (Invitrogen, Grand Island, NY). Modifications were carried out in aqueous solution buffered by 0.01 M NaH$_2$PO$_4$ with 0.01 M NaH$_2$CO$_3$, pH 8.2 for thirty minutes at room temperature. Reaction products were separated by gel filtration on Illustra NAP-5 columns (GE Healthcare, Piscataway, New Jersey) to remove unbound dye from labeled protein. CTxB (Invitrogen) was biotylated and conjugated to Atto 655 in-house via similar methods, and both CTxB and f(Ab)$_1$ conjugation was also described previously (*Stone and Veatch, 2015*). f(Ab)$_1$ was also conjugated to silicon rhodamine (SiR) dye (Spirochrome, Switzerland) in conjunction with biotin by similar methods. Streptavidin (Invitrogen) and anti-mouse IgG2b (Jackson ImmunoResearch; RRID: AB_2338463) were also conjugated to either Alexa 532 (Invitrogen) or Atto 655 by similar methods. CTxB that was only biotinylated (without Atto 655 conjugation) was purchased directly from Invitrogen. The commonly used STORM dye Alexa 647 was not used in most cases due to issues associated with the presence of near-red fluorophores present in reactive dye stocks (*Stone and Veatch, 2014*).

## DNA constructs

Lyn-eGFP, PM-eGFP, and eGFP-GG plasmids (*Pyenta et al., 2001*) were a generous gift from Barbra Baird and David Holowka (Cornell University, Ithaca, NY) and were cloned using standard techniques to replace eGFP with mEos3.2. Plasmid DNA encoding mEos3.2 protein and YFP-TM anchor sequences were gifts from Akira Ono (University of Michigan, Ann Arbor, MI). The YFP and mEos3.2 tagged constructs used here are in Clontech N1 plasmid vector background (Clontech, Mountain View, CA). The clathrinHC-GFP and clathrinHC-mEos3.1 constructs (*Sochacki et al., 2014*) were gifts from Justin Taraska (National Heart, Lung, and Blood Institute, NIH, Bethesda, MD).

We also cloned the CD45 transmembrane domain (termed $CD45_{tm}$) with a small number of flanking amino acids de novo from the amino acid sequence for mouse CD45 isoform 1, UniParc identifier P06800–1. We included the HA membrane-targeting signal sequence and a FLAG tag (*Guan et al., 1992*) upstream of the construct on the N terminus to allow for efficient plasma membrane delivery and detection, respectively. The signal sequence is cleaved from the construct in the ER prior to trafficking to the plasma membrane. The CD45 transmembrane domain was cloned into the HA-FLAG tag plasmid using standard techniques.

Amino Acid sequence of $CD45_{tm}$ insert with upstream HA-FLAG tag, where the signal sequence is shown in italics and the FLAG tag is shown in bold:

N terminus-*MKTIIALSYIFCLVFA***DYKDDDDA**NESTNFNAKALIIFLVFLIIVTSIALLVVLYKIYDLRKKR-C terminus

## Cells and transfection

CH27 cells (RRID: CVCL_7178), a mouse B cell lymphoma-derived cell line, was used as a model system for B lymphocyte signaling through the BCR (*Haughton et al., 1986*). Cells were acquired from Neetu Gupta (Cleveland Clinic), and cell line identity was authenticated using several criteria. Surface expression of mouse IgM, which is a specific marker of mouse B lymphocytes, was confirmed through specific labeling with goat anti-mouse IgM f(ab)$_1$ fluorescent conjugates. Cell morphology was typical for the B lymphoma cell type. Growth rates were monitored for consistency over time and cells were not kept in passage for longer than 60 days. Cultures tested negative for mycoplasma contamination. CH27s do not appear on the list of commonly mis-identified cell lines maintained by the International Cell Line Authentication Committee. Cells were maintained in culture as described previously (*Stone and Veatch, 2015*). CH27 cells were transiently transfected by Lonza Nucleofector electroporation (Lonza, Basel, Switzerland) with electroporation program CA-137. Generally, 700,000 CH27 cells were transfected with 1 μg plasmid DNA, except for Lyn where 500,000 cells were transfected with 0.7 μg plasmid DNA to avoid cell death. Cells were grown overnight on glass bottom wells (MatTek Corporation, Ashland, MA) at 200,000 per well. A subset of CH27 cells adhere spontaneously to glass bottom wells via an unknown mechanism but adhesion is not, in our experience, potentiated by first coating wells with fibronectin. For GG expression, cells were grown overnight in flasks, harvested and spun down, washed by pelleting and re-suspending three times in media, and then plated on the same day as labeling and fixation to minimize coverslip-bound mEos3.2-GG because this construct is secreted from cells. For cholesterol depletion and addition, cells were incubated for 15 min at 37°C with indicated concentrations of MβCD or cholesterol loaded MβCD (Sigma) freshly dissolved in Balanced Salt Solution (BSS: 135 mM NaCl, 1 mM $MgCl_2$, 1.8 mM $CaCl_2$, 5.6 mM glucose, 20 mM HEPES, pH 7.4). Cells were chemically fixed with 4% paraformaldehyde and 0.1% glutaraldehyde in PBS buffer or 2% paraformaldehyde and 0.15% glutaraldehyde in half-strength PBS for 10 min at room temperature unless otherwise indicated, and fix was quenched by washing with 5 mg/mL BSA.

Primary B lymphocytes were purified from a C57BL/6 mouse (Jackson Laboratories; RRID:IMSR_JAX:000664) using a standard negative selection procedure. All experiments were performed in compliance with federal laws and institutional guidelines as approved by the University of Michigan Committee on Use and Care of Animals. Briefly, one mouse was sacrificed using $CO_2$ asphyxiation. Spleen and lymph nodes were harvested in the presence of DNase I and filtered through a 70 μm strainer. Cells were pelleted and resuspended in DMEM with 2% FBS, 10 mM HEPES, 50 IU/mL penicillin, 50 μg/mL streptomycin, and 0.2 mg/mL DNAse I. 5 μg/mL CD11c (clone N418, Biolegend; RRID: AB_313772) and 5 μg/mL CD43 (clone S7, BD Biosciences; RRID: AB_2255226) biotinylated antibodies were added to cells for 30 min on ice prior to red blood cell lysis with RBC lysis buffer

(0.14 M $NH_4Cl$ and 0.017 M Tris, pH 7.2) and washing by pelleting. Remaining cells were incubated with streptavidin MACS beads (Miltenyi Biotec) for 20 min on ice and non-B cells were removed using an Automacs (Miltenyi Biotec) on the DEPLETES protocol. Primary B cells were then put into a buffer recommended by Lonza: RPMI 1640 supplemented with 10% FCS, 2 mM glutamine, 50 µM 2-mercaptoethanol, and 50 µg/mL LPS for 24 hr. Electroporation was accomplished with the P4 Primary Nucleofector solution with electroporation program DI-100 (Lonza) using 600,000 cells with 0.6 µg plasmid DNA in each well. Cells were grown overnight in flasks, spun down and washed extensively in cell media, and then plated onto fibronectin plates for 2 hr prior to labeling with f(Ab)$_1$biotin Atto 655, clustering with streptavidin, and fixation as described above.

RBL-2H3 cells (ATCC CRL-2256; RRID: CVCL_0591), a rat basophilic leukemia-derived cell line, were obtained from Barbara Baird and David Holowka (Cornell University). Cell identity was authenticated by expression of the high-affinity receptor for IgE, FcεRI, which was confirmed by specific binding of fluorescent IgE conjugates to the surface of cells. Cells were checked for characteristic morphology (*Siraganian et al., 1982*), growth rates were monitored for consistency over time, and cells were not kept in passage for longer than 90 days. Cultures tested negative for mycoplasma contamination. RBL-2H3 cells do not appear on the list of commonly mis-identified cell lines maintained by the International Cell Line Authentication Committee. RBL-2H3 cells were maintained in minimum essential medium with L-glutamine and phenol red with 20% fetal bovine serum and 0.1% gentamycin at 37°C in 5% $CO_2$, as described previously (*Gosse et al., 2005*). RBL-2H3 cells were transiently transfected with membrane anchor probes using the protocol described above for CH27 cells, with electroporation program DS-138.

HeLa cells were obtained from Akira Ono (University of Michigan) and maintained in high-glucose (4 mg/ml) Dulbecco's Modified Eagle Medium with 5% fetal bovine serum and 1% pen strep at 37°C in 5% $CO_2$. HeLa cells were only used for experiments to demonstrate properties of the analytical methods used (*Figure 6*), where cell identity was not pivotal to the interpretation of results. Cells had morphology and adhesive qualities common to this cell line but were not subjected to additional authentication. HeLa cells were transiently transfected using the protocol described above for CH27 cells, with electroporation program CN-114.

## Antibodies and labeling

For BCR experiments, goat anti-mouse IgM (Jackson ImmunoResearch, West Grove, PA; RRID: AB_2338477) f(Ab)$_1$ fragments conjugated to both fluorophores and biotin were used to label endogenous BCR in the plasma membrane. For fixed cell experiments, cells were stained with 5 µg/ml f(Ab)$_1$ conjugated to Atto 655 for 10 min in BSS followed by extensive washing prior to clustering with 1 µg/mL streptavidin in BSS prior to chemical fixation. Cross-correlations and images of fixed CH27 cells are shown for cells stimulated with antigen for 5 min prior to fixation unless otherwise noted. Primary cells were stimulated for 1 min prior to chemical fixation. For live cell experiments, cells were stained with 5 µg/ml f(Ab)$_1$ conjugated to SiR and biotin in BSS for 10 min. Images and data from live cells were acquired between 0 and 6 min after streptavidin was added at 1 µg/mL.

For clustered CTxB experiments, labeling of CTxB clusters was accomplished in one of two ways. Plasma membrane GM1 was bound with biotinylated CTxB at a concentration of 1 µg/mL for 10 min at room temperature in BSS. Cells were then washed extensively before adding 50 µg/mL streptavidin conjugated to Atto 655 for 10 min prior to chemical fixation. In some cases, plasma membrane GM1 was bound with 0.5 µg/mL biotinylated CTxB conjugated to Atto 655 for 10 min at 37°C. B cells were then washed with 37°C BSS buffer before clustering CTxB with 0.1 mg/mL streptavidin for 5 min at room temperature prior to chemical fixation. These two labeling methods produced equivalent results within error.

For clustered TM experiments, TM bearing an extracellular YFP tag was transfected into CH27 cells and subsequently clustered with 13 µg/mL anti-GFP rabbit IgG conjugated to biotin (ThermoFisher; RRID: AB_1090214) for 30 min at room temperature in BSS. Cells were then washed with BSS buffer and stained for 10 min with 100 µg/mL streptavidin conjugated to either Atto 655 when TM clusers were imaged in conjunction with mEos3.2 or Alexa 532 when TM clusters were imaged in conjunction with Atto 655.

For phosphotyrosine detection, fixed cells were permeablized with 0.1% Triton-X 100 in block buffer (PBS with 3% fish gelatin with 2 mg/mL BSA) and labeled with a 1:1000 dilution of anti-phosphotyrosine clone 4G10 primary antibody (Millipore, RRID:AB_916370) in block buffer for 1 hr. Cells

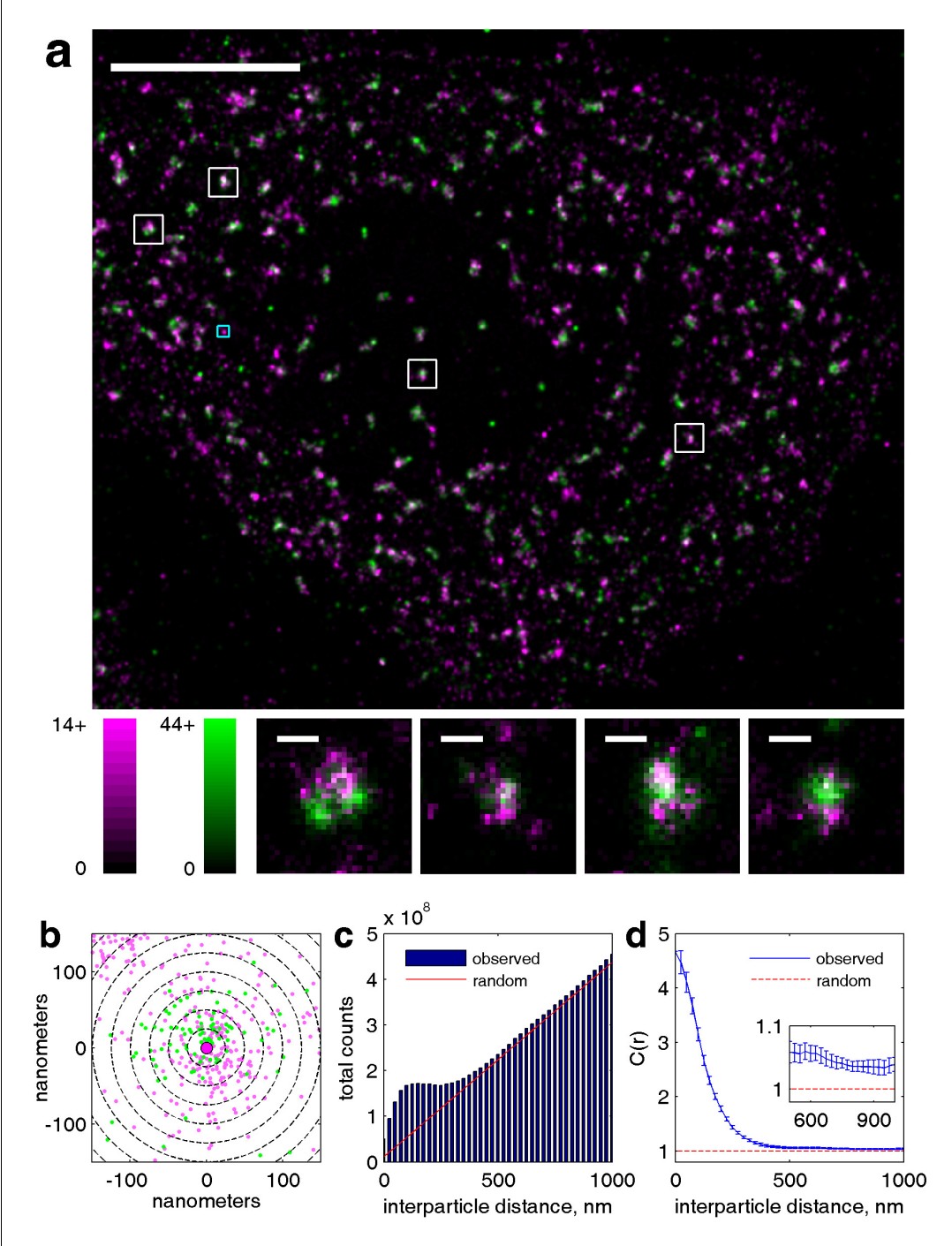

**Figure 6.** Our cross-correlation methodology applied to doubly labeled clathrin. (**a**) Two-color super-resolution image of a HeLa cell expressing two distinct labeled clathrin heavy chain constructs is shown. Clathrin heavy chains associate strongly in clathrin coated pits, and thus serve as an example of highly correlated co-clustered objects. Individual clathrin coated pits are shown below in the smaller images. Scalebar in large image is 5 μm, scale-bars in small images are 200 nm. (**b**) Zoom in of cyan box shown in large image, where position of points are plotted around an arbitrarily chosen central magenta localization. Dotted lines show the spatial bins used for calculating the cross-correlation function, where the number of green localizations within each bin are counted. In the complete cross-correlation these counts would also be summed over all magenta localizations. (**c**) Raw histograms of interparticle distances containing all pairs of particles localized within this cell. The red line shows the expected number of pairs in each spatial bin given a random distribution of both magenta and green localizations. The raw histogram is normalized by this curve to yield c(r). (**d**) Cross-correlation derived from localizations within this cell. Magnitude of the correlation indicates fold increase of pairs detected at the specified inter-particle distance

*Figure 6 continued on next page*

*Figure 6 continued*

compared to a random distribution. The expected value of the cross-correlation given a random co-distribution is equal to one due to the normalization. Error bounds shown are $dC_2(r)$ and are estimated from the statistics and resolution of the image as defined in *Equation 3*.

The following figure supplements are available for figure 6:

**Figure supplement 1.** The cross-correlation function detects deviations in the co-distribution of localizations from random.

**Figure supplement 2.** The amplitude of the cross-correlation reflects differences in enrichment magnitude and interaction strength.

**Figure supplement 3.** Cross-correlations detect co-clustering even when there is a low surface density of labeled molecules.

**Figure supplement 4.** Membrane topology gives rise to long-range structure in cross-correlations, but can be removed by careful selection of regions of interest (ROI).

**Figure supplement 5.** Estimation of the variance associated with a cross-correlation measured on a single super-resolution fluorescence localization image.

were washed extensively before adding the secondary antibody, goat anti-mouse IgG 2b subtype specific (Jackson ImmunoResearch; RRID: AB_2338463). The secondary antibody was conjugated to either Atto 655 (when observing TM clusters) or Alexa 532 (when observing BCR and CTxB clusters) prior to use in labeling.

For CD45 detection, anti-mouse CD45R (B220) primary antibody clone RA3-6B2 conjugated directly to Alexa 532 was used (eBiosciences, San Diego, CA; RRID: AB_467253). After cells were fixed, 1 µg/mL antibody was allowed to bind to endogenous CD45 for 2 hr at room temperature in block buffer before washing to remove unbound antibody.

For CD45tm detection, B cells transfected with FLAG-CD45tm were chemically fixed and then stained with 20 µg/mL mouse monoclonal anti-FLAG M1 (Sigma; RRID:AB_439712) in block buffer for 1 hr at 37°C. Cells were then washed extensively before labeling with 10 µg/mL goat anti-mouse IgG 2b conjugated to Alexa 532 for one hour at 37°C in block buffer.

For endogenous Lyn and phosphorylated Lyn detection, chemically fixed B cells were permeablized with 0.1% Triton-X 100 in block solution following clustering of CTxB-biotin with streptavidin-Atto 655. Samples were then incubated with either a 1:50 dilution of anti-Lyn primary antibody (rabbit polyclonal anti-Lyn IgG clone 44; Santa Cruz Biotech; RRID:AB_2281450) or a 1:100 dilution of anti-phospho-Lyn primary antibody (rabbit monoclonal IgG anti-pY397 Lyn clone EP503Y; Abcam; RRID:AB_776106) for 1 hr at room temperature. Samples were washed extensively in block buffer and then incubated with a 1:1000 dilution of Alexa Fluor 532 conjugated goat anti-rabbit secondary antibody (goat polyclonal anti-rabbit IgG (H+L); ThermoFisher; RRID: AB_10374433). Samples were washed in block to remove unbound antibody.

For BCR and CTxB cross correlation, biotinylated CTxB was clustered by streptavidin conjugated to Atto 655 and cells were fixed prior to labeling BCR with anti-IgM f(Ab)$_1$ fragments conjugated to Alexa 532. Samples were washed in block to remove unbound antibody.

Two-color super-resolution images of clathrin coated pits were obtained by co-expressing two alternatively labeled clathrin heavy chain (HC) constructs, clathrinHC-GFP and clathrinHC-mEos3.1, in HeLa cells. One million HeLa cells were co-transfected with 0.75 µg clathrinHC-mEos3.1 as well as 0.75 µg clathrinHC-GFP. Cells were fixed and membranes were permeablized as above, blocked in 2% BSA, and GFP was labeled with a biotinylated anti-GFP primary antibody (ThermoFisher; RRID: AB_1090214). Subsequently, cells were washed extensively and streptavidin bound to Alexa 647 (Invitrogen) was added to label clathrinHC-GFP for imaging.

## TIRF microscopy

Imaging was performed on an Olympus IX81-XDC inverted microscope with a cellTIRF module, a 100X UAPO TIRF objective (NA = 1.49), and active Z-drift correction (ZDC) (Olympus America, Center Valley, PA) as described in previous work (*Stone and Veatch, 2014*, *2015*). Images were acquired on an iXon-897 EMCCD camera (Andor, South Windsor, CT). Excitation of Atto 655 was

accomplished using a 647 nm solid state laser (OBIS, 100 mW, Coherent, Santa Clara, CA) when imaged in conjunction with mEos3.2, or a 640 nm diode laser (CUBE 640-75FP, Coherent) when imaged in conjuction with Alexa 532. Excitation of mEos3.2 constructs was accomplished using a 561 nm solid state laser (Sapphire 561 LP, Coherent). Photoactivation of mEos3.2 was accomplished with a 405 nm diode laser (CUBE 405-50FP, Coherent). Excitation of Alexa 532 was accomplished with a 532 nm diode-pumped solid-state laser (Samba 532–150 CW, Cobolt, San Jose, CA). Laser intensities were adjusted such that single fluorophores could be distinguished in individual images, and were generally between 5 kW/cm$^2$ and 20 kW/cm$^2$. Excitation and emission was filtered using a LF405/488/561/647 quadband cube (TRF89902, Chroma, Bellows Falls, VT) or a 532/640 dualband cube (TRF59907, Chroma). Emission was split into two channels using a DV2 emission splitting system (Photometrics, Tuscon, AZ) using a T640lpxr dichroic mirror to separate emission, ET605/52m to filter near-red emission, and ET700/75m to filter far-red emission (Chroma). Chemically fixed samples with Atto 655 and mEos3.2 were imaged in a buffer suitable for STORM and PALM microscopy: 30 mM Tris, 9 mg/ml glucose, 100 mM NaCl, 5 mM KCl, 1 mM KCl, 1 mM MgCl$_2$, 1.8 mM CaCl$_2$, 10 mM glutathione, 8 µg/ml catalase, 100 µg/ml glucose oxidase, pH 8.5. Live samples were imaged with the same buffer except with 200 µg/ml catalase at pH 8, which is more suitable for live cells since it has enhanced reactive oxygen species scavenging and the pH is closer to physiological pH. Fixed samples with Atto 655 and Alexa 532 or with Alexa 647 and mEos3.1 were imaged in a buffer more suitable for oxazine and rhodamine dyes (*Heilemann et al., 2009*): 50 mM Tris, 100 mg/mL glucose, 10 mM NaCl, 100 mM 2-mercaptoethanol, 50 µg/ml glucose oxidase, 200 µg/ml catalase, pH 8. In some cases, glucose oxidase concentration was lowered or it was omitted from the buffer entirely in order to optimize the photoswitching rates of Atto 655 and Alexa 532. Live cells were imaged at approximately 45 frames per second with an exposure time of 20 milliseconds, and the exposure time for fixed cells varied between 20 and 50 milliseconds.

## Super-resolution image reconstruction

Single molecule fluorescent events were localized by fitting local maxima in background subtracted images to Gaussian functions using standard methods. The ensemble of peaks was then culled to remove outliers in brightness, size, and localization error using in-house MATLAB software (*Veatch et al., 2012*). For live cells, single molecules were localized in raw live cell movies with the ImageJ plugin ThunderSTORM (*Ovesný et al., 2014*), using weighted least-squares fitting of an integrated Gaussian PSF with multi-emitter fitting analysis enabled to detect up to two single molecules within a diffraction-limited area. Localization data were then exported to our in-house MATLAB software for culling and successive post-processing steps (*Veatch et al., 2012*). Localizations in the near-red emission channel were registered with the far-red emission channel using a registration technique published previously (*Churchman et al., 2005*) and previously used by our group (*Stone and Veatch, 2014*, *2015*). Stage drift correction was performed every 500 frames by finding the maximum in the 2D cross correlation produced by all localizations between successive groups of frames. Super-resolution localizations were used to reconstruct super-resolved images after correcting for stage drift and channel registration by incrementing the intensity of pixels at positions corresponding to localized single molecules. The super-resolved images have an arbitrary pixel size of 25 nm, and the original images have a pixel size of 160 nm, corresponding to the pixel size of the EMCCD camera. For the purposes of display, localizations were grouped such that probes observed within a small (typically 80 nm) radius in sequential frames were merged and counted as a single localization. Note that this grouping correction does not account for multiple observations of the probe imaged at different times, for example as a result of reversible activation. Histograms of localized positions were blurred as described in figure captions and image contrast was adjusted for display purposes. The resolution of particle localization was close to 30 nm for all probes, determined by correlation-based methods as detailed previously (*Veatch et al., 2012*). This resolution is larger than the localization precision of the Gaussian fits because it includes contributions from other sources of error (e.g. from stage drift).

## Cross-correlation analysis in chemically fixed cells

Regions containing cells were masked by a user-defined region of interest (ROI), and cross correlations were computed from these regions using methodology described previously (*Sengupta et al.,*

*2011*; *Veatch et al., 2012*; *Stone and Veatch, 2015*) and summarized here. Cross-correlation functions report on the enrichment or depletion of distinguishable probes with respect to one another, normalized by a random co-distribution of probes. Thus, the magnitude of the cross-correlation yields information about interactions of labeled objects with one another that may cause their co-distributions to deviate from a random co-distribution. This methodology is demonstrated in *Figure 6* using imaging of dually-labeled clathrin coated pits as an example. Clathrin coated pits were imaged in HeLa cells transiently expressing two distinct labeled clathrin heavy chain proteins, one conjugated to mEos3.1 (green) and a second conjugated to GFP that is antibody labeled with Alexa 647 (magenta). Clathrin was chosen for this demonstration because a large number of individual clathrin proteins assemble within clathrin coated pits, which are sparsely distributed within the cell, therefore their co-localization is easily identified when viewing the reconstructed image. A reconstructed image of a HeLa cell showing the distributions of super-resolved localizations arising from both clathrin constructs is shown in *Figure 6a*.

The correlation function can be assembled by tabulating the pair-wise distances between distinguishable probes localized within a masked image, then binning these separation distances to produce the average number of pairs separated by distances between r and r+$\Delta$r, where $\Delta$r is usually 25 nm in our measurements. The point distribution of distinguishable probes surrounding an example magenta probe is shown in *Figure 6b*. In this case, there are many more green points located at small separation distances from the example magenta point than at large separation distances because the magenta point chosen was located at the center of a clathrin coated structure. These pairwise distances are collected within the radial bins given by the dotted lines. The complete correlation function tabulates these separation distances around all magenta probes in the image. *Figure 6c* shows the histogram describing the distribution of separation distances for all pairs from the cell shown in *Figure 6a*. In general, the number of pairs in each bin increases linearly with increasing radius for large separation distances because the area corresponding to each bin also increases linearly, as A $\approx$ 2$\pi$r$\Delta$r. These histograms are then normalized by the total number of observations divided by the area of the cell and multiplied by the area of the bins. This normalization is equivalent to the number of observations expected in each bin given a random distribution of pairs across all bins, and is shown as a red line in *Figure 6c*. Importantly, this normalization simply accounts for variation in expression level between cells (*Figure 2—figure supplement 3*) and corrects for boundary effects that arise due to the finite extent and shape of the ROI. The cross-correlation function can be equivalently calculated from reconstructed images of all localizations using fast Fourier transforms, as has been described previously (*Veatch et al., 2012*; *Stone and Veatch, 2015*). In this case, a two dimensional cross-correlation is tabulated, C(r, $\theta$), and then C(r) is obtained by averaging over angles. Generally, C(r) is tabulated from ungrouped images, meaning that localizations detected within a small radius in sequential frames are counted independently. Ungrouped images are used because cross-correlation functions are not impacted by probe over-counting (*Veatch et al., 2012*) and this reduces possible errors introduced by the grouping technique. The properly normalized cross-correlation function for this cell is shown in *Figure 6d*.

Cross-correlation functions only indicate significant correlations when the spatial distribution of one probe influences the spatial distribution of the second probe, even when one or both of the probes are clustered themselves. This effect is demonstrated in *Figure 6—figure supplement 1*. Error bars on this curve are estimated using the variance within the radial average of the two dimensional C(r, $\theta$), the average lateral resolution of the measurement, and the numbers of probes imaged in each channel, as described in detail below.

As expected, the cross-correlation function tabulated from the image shown in *Figure 6a* indicates that probes are highly co-localized, where the co-localized density within the first spatial bin (r < 25 nm) is five times higher than randomly co-distributed probes. In this case, C(r) $\approx$ 1 for separation distances much larger than the size of individual clathrin structures, meaning that the pits themselves are roughly randomly distributed on the cell surface. C(r) is slightly larger than one even at separation distances approaching 1 µm because clathrin structures are more densely localized on the edge of this cell than towards the center. In some instances, we subtract this long distance offset in order to remove long range contributions to C(r) which are not currently under investigation. The vast majority of cross-correlation functions reported in the main text have much smaller amplitudes than the one shown in this example. This is because co-localization is much weaker and/or domains

are more numerous. Examples of single cell correlation functions for various probes that co-localize with BCR clusters along with reconstructed images are shown in *Figure 6—figure supplement 2*.

A distinct advantage of this cross-correlation function approach is that it involves averaging over multiple domains within an image, and can be further averaged over images. This makes it possible to quantify co-localization that is far too weak or under-sampled to be apparent from visual inspection of images. This is demonstrated in *Figure 6—figure supplement 3*, which shows a simulated case where the same weak co-distribution of probes is sampled to varying degrees. When the spatial distributions are well-sampled, then co-localization is easily apparent both visually in the image and quantitatively in the tabulated correlation function. When spatial sampling is low, co-localization is no longer apparent in images, and in fact probes can appear anti-correlated because sampling is so sparse that localizations are unlikely to be overlapping. However, cross-correlation functions can still detect co-localization in many cases, although reduced sampling decreases the signal-to-noise.

ROIs are chosen so that only flat regions of the cell surface are analyzed, which in some cases meant that regions of the cell interior were not included in the ROI when the membrane lifts from the TIR field and membrane components are no longer visualized (*Figure 6—figure supplement 4*). When included in the ROI, regions of membrane topology produced correlations that extend to large radii (>200 nm) in tabulated cross-correlation functions, as shown in *Figure 6—figure supplement 4a–b*. This is because both probes are necessarily absent in regions where the membrane has lifted from the glass surface, which makes probes correlated. The normalization of the cross-correlation function properly accounts for complex regions of interest. Significant efforts were made to minimize the impact of membrane topology, but in some cases this was complicated by low spatial sampling of labeled proteins and peptides. Especially in cases where spatial sampling is low, user-defined ROI have the potential to introduce systematic bias that could impact cross-correlation results. In some cases, cells were analyzed without user knowledge of the sample condition, and results were indistinguishable within noise. We also found little user-to-user variation in cross-correlations determined from single cells or averaged over a population (*Figure 6—figure supplement 4c–d*).

## Over-counting and estimating protein/peptide surface densities

One major limitation of the super-resolution methods and probes used here is that it is not possible to simply distinguish multiple observations of the same labeled molecule from a small aggregate of labeled molecules. However, it is possible to estimate the average surface density of labeled molecules for cases where probe blinking follows Poisson statistics and where probes are nearly randomly distributed (*Veatch et al., 2012*). This is accomplished by fitting a Gaussian function with standard deviation σ and amplitude A to the autocorrelation function tabulated from a single color image. This single color image is reconstructed from grouped localization data, meaning that localizations detected within a small radius (80nm) in sequential frames are counted as a single localization. Grouping sequential localizations produces images with sampling that better approximates Poison statistics, since localizations are less correlated in time. When labeled proteins are randomly or nearly randomly distributed in space, the area under of the autocorrelation function is inversely proportional to the surface density of labeled proteins according to:

$$\rho = \frac{1}{2\pi\sigma^2}\frac{1}{A} \tag{1}$$

We expect this to be an accurate estimate of surface density for the majority of mEos3.2 conjugated peptides used in this study, since they are expected to be only subtly self-clustered within the membrane. This estimate will be less accurate for the case of proteins with higher-order structure including extended clusters, such as clustered BCR and CTxB where this density is likely better interpreted as the density of clusters, not individual proteins. We can estimate the average number of times each independent protein or peptide structure is sampled by comparing the average density of localizations to the average surface density of labeled proteins or peptides determined using *Equation 1*. For the localization data presented in this study, we generally find that independent proteins and peptides are observed between 10 and 50 times over the 5000–10,000 raw acquisition frames imaged. While the cross-correlation obtained between reconstructed images of two different probes is not adversely affected by over-counting, over-counting does impact the observed variance, as described below.

## Variance of cross-correlation measurements

Estimating error bounds on individual fixed cell measurements is complicated by the presence of over-counting of single labeled proteins in combination with finite localization precision. In the absence of these two effects, the variance in C(r) can be simply calculated using Poisson statistics to describe the probability of detecting a certain number of average pairs within some specified area given the cross-correlations observed. This strategy has been applied to estimate error on single live cell cross-correlations (*Stone and Veatch, 2015*), but it depends strongly on the average densities of the labeled proteins present. In fixed cells, these numbers can be only estimated due to over-counting as described above. Instead, the variance is estimated by calculating dC(r, <θ>), the standard deviation of the mean obtained when averaging the 2D cross-correlation function C(r, θ) over angles to extract C(r) as outlined in *Figure 6—figure supplement 5* and described below.

In the limit of resolution much smaller than the pixel size, dC(r,<θ>) accurately reproduces the variance obtained by observing many replicates of a simulation where two distinguishable probes are distributed randomly, as shown in *Figure 6—figure supplement 5a*. When the image resolution is on the order of or larger than the pixel size, then there is smoothing of the image and the resulting C(r, θ). In this case, it is not appropriate to simply tabulate the standard error of the mean of pixel values falling within a separation distance range between r and r+Δr because neighboring pixels in the two-dimensional C(r, θ) are correlated. When the localization precision is known, this effect can be simply corrected using a multiplicative factor that only depends on the localization precision, $\sigma_{PSF}$, which is the standard deviation of the super-resolved point spread function (PSF):

$$dC_1(r) = (1 + (\sqrt{2}\sigma_{PSF}/\Delta r)) \times \left(1 + e^{-r^2/4\sigma_{PSF}^2}\right) \times dC(r, \langle\theta\rangle) \tag{2}$$

In all instances presented here, $\sigma_{PSF}$ is taken to be 30 nm for the sake of this calculation. At radii much larger than $\sigma_{PSF}$, this factor simply corrects for the fact that correlated pixels in C(r, θ) are contributing to the average over angles, so the number of independent measurements is less than the number of pixels contributing to the average. At short radii, blurring over the super-resolved point spread function also decreases the amplitude of correlations directly, so there is additional under-estimation of variance by the simple angular average method. This correction factor is applied to simulations of blurred randomly distributed points in *Figure 6—figure supplement 5b*.

This multiplicative correction factor of *Equation 2* over-estimates the error when the super-resolved point spread function is not well sampled. This under-sampling introduces variance that should not be amplified by the correction factor shown above. This can be corrected further by subtracting a term that depends on the number of observations of each single color label ($N_1$ and $N_2$), the number of labeled proteins in the image ($n_1$ and $n_2$), and the size of the super-resolved point spread function ($\sigma_{PSF}$):

$$dC_2(r) = dC_1(r) - \sqrt{\frac{4\pi\sigma_{PSF}^2}{\Delta r^2}\left(\frac{N_1^2}{n_1} + \frac{N_2^2}{n_2}\right)^{-1}} \times \left(1 + 4e^{-r^2/4\sigma_{PSF}^2}\right) \tag{3}$$

The pre-factor on this correction term represents how well the area occupied by all probes ($4\pi\sigma_{PSF}^2 n$) is sampled by pairs of localizations of that color ($N^2$). This term becomes negligible when there are many localizations per probe in either channel, as is typical in the fixed cell measurements presented here. For this reason it does not contribute significantly to the results presented. The number of labeled proteins in each channel (n1 and n2) is estimated by fitting the autocorrelation to extract the density of independent objects using *Equation 1* above and then multiplying this number by the area of the region of interest. *Figure 6—figure supplement 5c* shows that $dC_1(r)$ is sufficient to describe the simulation-to-simulation variation when objects are sampled 20 times, which is typical of the images investigated in this work. When sampling is lower, this correction is needed to more accurately estimate the the simulation-to-simulation variation as presented in *Figure 6—figure supplement 5d*. The Gaussian shape of this correction is estimated from simulations and may not apply in all contexts.

In the majority of cases where correlation functions are presented within figures, the values plotted are averaged together across cells of the same treatment and condition to obtain the average correlation function, and the error bars represent the standard error of the mean between cells. The number of cells going into each average is shown in figure captions and figure supplements and,

with the exception of the primary cell experiments, includes at least two biological replicates where samples were prepared for imaging on separate days. In all cases we have examined closely, the average error estimated from a single measurement of the cross-correlation function is close to the width of the distribution of single cell cross-correlation values, indicating that the observed variation is dominated by counting statistics and not more systematic differences between cells within the population. Examples demonstrating this point are shown in *Figure 1—figure supplement 5* and *Figure 2—figure supplement 2*.

## Steady-state cross-correlation and step-size analysis in live cells

Cross-correlations from live cells were calculated as described previously (*Stone and Veatch, 2015*), where the time evolution of the cross correlation was used to better specify the instantaneous cross-correlation. In brief, cross-correlation functions were computed on a frame-by frame basis from localizations in each channel that occurred in the same frame or in frames separated by a time delay τ. Cross-correlations between frames with time separation of up to 50 frames (0 s < τ < 1 s) did not decay significantly (*Figure 2—figure supplement 6*) and were therefore averaged to obtain a steady-state cross-correlation for data collected in a time window between 0 and 6 min after clustering with streptavidin. Long-range gradients in labeling density arise in live-cell data because labeled molecules continually diffuse onto the ventral membrane from the dorsal membrane during the imaging experiment. The dorsal membrane is outside the reach of TIRF illumination and away from the high laser power that both converts probes to a fluorescent 'off' state and slowly bleaches them. Therefore, probes near the edges of the cell footprint are more likely to reside in a fluorescent 'on' state, and as a result these areas are more densely sampled. To compensate for the effects of this long-range structure on our measurement, we normalize steady-state cross-correlations by the cross-correlation function of the masked average images from each channel which are first convoluted with a two-dimensional Gaussian function with σ = 1 μm. This treatment filters structure larger than 1 μm in size from the steady-state cross-correlation function.

For step-size analysis, single molecule trajectories were constructed from super-resolution localizations using a tracking algorithm that searches for localizations within 500 nm in subsequent frames and terminates ambiguous trajectories (*Shelby et al., 2013*). The step size distribution for BCR-correlated probes is calculated by finding all instances of probe localization within 100 nm of a simultaneous BCR localization, and comparing that position to the location of the probe in immediately preceding and subsequent frames. These step sizes were compiled over tens of thousands of frames from multiple single-cell experiments.

## Calcium measurements

For measurements of calcium mobilization following BCR clustering and activation, 5 million CH27 cells were loaded with 2 μg/mL Fluo-4 AM (Invitrogen) for 5 min at room temperature in 1 mL BSS buffer with 0.25 mM sulfinpyrazone. The cell suspension was subsequently diluted to a final volume of 15 mL with BSS buffer and incubated for 30 min at 37°C to allow for dye loading. 700,000 cells in 1.8 mL BSS buffer were then treated with either methyl-β-cyclodextrin (MβCD), MβCD loaded with cholesterol (Sigma), or left untreated at 37°C for 15 min. The concentrations of both MβCD+cholessterol and MBCD were determined by the molecular weight of MBCD alone, 1310 Da. For each treatment condition, cells were then spun down and resuspended in 1 mL of calcium-free PBS with 0.25 mM sulfinpyrazone. Cells were spun down again and resuspended in 400 μL PBS. Approximately 300,000 cells were loaded into individual wells of a black 96 well plate. Fluo-4 was visualized on a fluorescence plate reader (Omega; BMG Labtech, Ortenberg, Germany) using excitation centered at 485 nm and emission centered at 520 nm. Cells were stimulated by addition of f(Ab)$_2$ goat anti-mouse IgM (Jackson Immunoresearch; RRID:AB_2338469) to a final concentration of 3 μg/mL. Average calcium mobilization curves were generated from 2–4 wells per treatment condition. Baseline drift was corrected by fitting a line to the Fluo-4 fluorescence trace prior to antigen addition and dividing the entire fluorescence trace by this baseline. Baseline-corrected fluorescence traces therefore reflect the fold increase in signal compared to spontaneous calcium release and fluorescence background. Baseline-corrected curves were then integrated over a two-minute window after antigen addition that captured the peak calcium response, as shown in *Figure 5—figure supplement 2*.

For calcium measurements with CTxB clustering, adherent CH27 cells were labeled with biotinylated CTxB in the same manner as super-resolution imaging measurements, described above, and then loaded with 0.4 µg/ml Fluo-4 AM in BSS buffer at 37°C for 30 min. Cells were washed and imaged in BSS buffer at room temperature at 10x magnification using a FITC filter set with epifluorescence excitation. Cells were imaged every 0.2 s for 1 min before and 8 min after addition of 50 µg/mL streptavidin. Data were recorded using a Neo sCMOS camera (Andor, South Windsor, CT). After data acquisition, Fluo-4 intensity traces for individual cells were tabulated through an automated image processing algorithm that localized cells and tracked pixel intensities corresponding to individual cells before and after stimulation. Raw intensity traces are shown in *Figure 3—figure supplement 1* and include the non-zero offset of the camera.

## Western blots

Western blots were performed on CH27 cell lysates that probed protein tyrosine phosphorylation following binding and clustering of CTxB or clustering of BCR. One million cells at a concentration of 2 million cells/mL were used for each sample. For samples where BCR was clustered, 10 µg/mL f (Ab)$_2$ goat anti-mouse IgM (Jackson Immunoresearch; RRID:AB_2338469) was added to cells for 2 min prior to cell lysis. For samples were CTxB was bound, cells were incubated with 10 µg/mL CTxB biotin for 10 min, followed either by cell lysis or CTxB clustering by incubation with 100 µg/mL streptavidin for an additional 2 min prior to lysis. Cells were lysed on ice for 20 min with shaking in 1X RIPA buffer containing 1X Halt Phosphatase Inhibitor Cocktail, 4 mM EDTA, and 1X solution of cOmplete Mini protease inhibitor tablet. Cell lysates were spun down for 15 min at 16,000g and 4°C, and the supernatant was collected. Lysates were flash frozen in liquid nitrogen and stored at −20°C. Samples were run on SDS PAGE gels with 10% acrylamide, and gels were transferred using the iBlot Dry Blotting System (ThermoFisher) as per manufacturers recommendations. Phosphotyrosine was detected by incubating blots with a 1:2500 dilution of anti-phosphotyrosine 4G10 Platinum primary mouse antibody (Millipore, RRID:AB_916370) overnight at 4°C. Blots were then incubated in a 1:1000 solution of horseradish peroxidase-conjugated secondary antibody (goat anti-mouse IgG, Fcγ subclass 2b specific; Jackson ImmunoResearch; RRID:AB_2338515) for 2 hr at room temperature. Actin labeling was used as a loading control, and blots were stripped and re-probed with a 1:1000 solution of rabbit polyclonal anti-actin primary antibody (Cytoskeleton; RRID:AB_10708070) followed by a 1:1000 solution of horseradish peroxidase-conjugated secondary antibody (goat anti-rabbit IgG; Jackson ImmunoResearch; RRID:AB_2307391). Chemiluminescence was captured using a GelDoc system. Blot band intensity was analyzed in MATLAB. Total band intensities were summed within user-defined regions after subtracting the average background intensity estimated from unused lanes. Total band intensities were normalized by corresponding actin band intensities for each lane.

## Plasma membrane vesicle isolation and measurement

For probe partitioning measurements (*Figure 1—figure supplement 2*), giant plasma membrane vesicles (GPMVs) were made from adherent rat basophilic leukemia cells (RBL-2H3, ATCC CRL-2256; RRID: CVCL_0591) using established protocols (*Baumgart et al., 2007*; *Veatch et al., 2008*; *Zhao et al., 2013*; *Gray et al., 2013*) with minor modifications. Prior to GPMV isolation, adherent cells were labeled with either 2 µg/mL CTxB conjugated to Alexa 647 (Invitrogen) for 10 min at room temperature or 3 µg/mL DiD C$_{16}$ (Invitrogen) in 0.03% methanol for 10 min at room temperature. When both DiD and CTxB were imaged, CTxB conjugated to Alexa 555 (Invitrogen) was used. Cells were rinsed and incubated in a buffer containing dithiothreitol (DTT; 2 mM) and formaldehyde (25 mM) in the presence of calcium (2 mM) at 37°C for 2 hr with gentle rocking. GPMVs were harvested and imaged at low temperature between two coverslips on a home built temperature-controlled stage as described previously (*Veatch et al., 2008*; *Zhao et al., 2013*; *Gray et al., 2013*). Vesicles were imaged on a separate IX81 inverted microscope (Olympus) using epifluorescence illumination with a Cy3 filter set (Chroma) for CTxB Alexa 555 and a Cy5 filter-set (Chroma) for DiD C$_{16}$. Images were captured on a Neo SCMOS camera (Andor). The partitioning of eGFP-GG and YFP-TM were examined by imaging GPMVs harvested from cells transiently expressing these constructs using a GFP filter cube (Chroma). DiD C$_{16}$ was used as a phase marker (*Figure 1—figure supplement 2*).

To examine PM anchor phase partitioning, GPMVs were prepared from cells expressing PM-eGFP as described above except with 4 mM glutathione substituted for DTT as the reducing agent. Glutathione was used as a reducing agent in these measurements because it is not cell permeable and therefore is not expected to directly impact the palmitoylation state of the PM peptide, whereas some reducing agents have been found to perturb protein palmitoylation in GPMVs (*Leventhal et al., 2010*). We note that GPMVs prepared using glutathione have lower transition temperatures and a larger surface fraction of ordered phase than GPMVs prepared using DTT. Due to the low phase separation temperature of vesicles prepared in this manner, 6 µM hexadecanol was added to raise the phase separation temperature to about 1°C (*Machta et al., 2016*) so that phase separated vesicles could be observed. GPMVs were imaged as described above.

To examine how the surface fraction of ordered and disordered phases varies with acute cholesterol variation, adherent CH27 cells were first pre-treated with either 10 mM MβCD or 10 mM MβCD pre-complexed with cholesterol for 10 min. Cells were then labeled with 2 µg/ml DiI-C$_{12}$ (Invitrogen) in 0.02% methanol for 10 min at room temperature and GPMVs were prepared and imaged as described above using DTT as the reducing agent. Fewer vesicles were obtained in MβCD or MβCD-chol pretreated cells than in untreated cells, likely because treated cells were less adherent.

## Simulations of receptors, kinases, and phosphatases in a heterogeneous membrane

A conserved order parameter 2D Ising model was simulated on a 256 by 256 square lattice as described previously (*Machta et al., 2011*) with minor modifications. Briefly, components that prefer ordered or disordered regions are represented as pixels that have value of S = +1 and S = -1 respectively. The vast majority of +1 and −1 pixels represent unspecified membrane components (proteins and lipids). In addition, 50 pixels with values of +1 are classified as receptors, 100 pixels with values +1 are classified as kinases, and 100 pixels with values −1 are classified as phosphatases. Receptors are clustered by applying a strong attractive circular field ($\varphi^R$) at the center of the simulation frame that only acts on receptors. The final Hamiltonian is given by:

$$H = -\sum_{i,j} S_i S_j - \sum_i R_i \Phi_i^R$$

The first term sums over the four nearest neighbors (j) surrounding the pixel i and applies to all components. The second term only contributes when receptors occupy position i, where $R_i$=1, otherwise $R_i$=0. The receptor field $\Phi_i^R$ has a circular shape with a radius of 16 pixels (32 nm) and is centered in a simulation box with periodic boundary conditions. When an ordered domain is stabilized in the absence of receptor clustering, a similar Hamiltonian is used with an applied field that is felt by all membrane components. In this case:

$$H = -\sum_{i,j} S_i S_j - \sum_i S_i \Phi_i^D$$

The domain field $\Phi_i^D$ has a circular shape with a radius of either 24 pixels (~50 nm) or 48 pixels (~100 nm) and is centered in a simulation box with periodic boundary conditions. The magnitude of this field was chosen to be equal to a single interaction between components, which is one in these units. This magnitude is sufficient to stabilize a robust domain containing ordered components but does not restrict the motions of individual components within the domain.

At each update, two random pixels are chosen, the energy cost or gain for exchanging the two pixels is calculated, and the move is either accepted or rejected using a Monte Carlo algorithm that maintains detailed balance. If the resulting configuration is lower or equal in energy, the exchange is always accepted. If the energy is raised, the exchange is accepted stochastically with probability exp($-\beta\Delta$H) where $\beta$ is the inverse temperature and $\Delta$H is the change in energy between initial and final states. In this scheme, the critical point occurs at T$_C$ = 2/ln(1+sqrt(2)). All simulations were run at T = 1.05 × T$_C$. One pixel is chosen to represent a 2 nm by 2 nm patch of membrane, so that the correlation length varies with temperature in simulations with equal fractions of ordered and disordered components as observed in experimental observations in isolated plasma membrane vesicles (*Veatch et al., 2008*). Most simulations were run such that there were an equal fraction of ordered and disordered unspecified membrane components. In some cases, the fraction of unspecified

membrane compositions assigned to be ordered was varied, as indicated in Figure captions. Uniform simulations were run by setting all unspecified membrane components to be disordered.

One sweep corresponds to the option to exchange each of the pixels on average twice ($256^2$ pixel swaps are proposed). All simulations are initially run using non-local exchanges to decrease equilibration times. For simulations recording receptor phosphorylation state, exchanges were then restricted to nearest neighbors in order to better mimic diffusive dynamics. Simulation sweeps are converted to time assuming a diffusion coefficient of roughly 4 µm²/s, with one sweep corresponding to roughly 1 µs. Most simulations were recorded for 1000 sweeps which corresponds to roughly 1 s.

If a move is accepted that places a receptor neighboring a kinase, then the receptor is phosphorylated at a low probability (0.1%). If a move is accepted that places a receptor neighboring a phosphatase, then the receptor is dephosphorylated at a high probability (100%). These probabilities are chosen to produce a low level of phosphorylation in simulations that contain an equal number of kinases and phosphatases with unclustered receptors. Higher probability of dephosphorylation is physiologically relevant because phosphatases such as CD45 are expressed in the plasma membranes of lymphocytes at several-fold higher densities than Src kinases (e.g. T cells express between 100,000 and 500,000 CD45 molecules and between 40,000 and 120,000 Lck molecules per cell (*Olszowy et al., 1995*; *Hui and Vale, 2014*). In some simulations, receptors have kinase behavior when they are phosphorylated. In this case, a move that places a phosphorylated receptor next to a second receptor results in the second receptor becoming phosphorylated at a low probability (0.1%).

To mimic the experimental limitation of finite lateral resolution, cross-correlation functions between receptors and membrane components were also tabulated from simulation snapshots that were first filtered with a Gaussian shaped point spread function with the indicated width. This is equivalent to convolving the raw two dimensional $C(r, \theta)$ with the autocorrelation of the point spread function $g_{PSF}(r)$ (*Veatch et al., 2012*).

All analyses were carried out in MATLAB (The MathWorks, Natick, MA; RRID: SCR_001622). Plasmids and reagents can be obtained via request of the corresponding author.

## Acknowledgements

We thank Jing Wu for assistance with some experiments and Akira Ono, Johnathan Grover, Barbara Baird, David Holowka, and Justin Taraska for supplying DNA constructs. We thank Neetu Gupta for CH27 cells, Barbra Baird and David Holokwa for RBL-2H3 cells, Akira Ono for HeLa cells, and Irina Grigorova for primary mouse B cells. We additionally thank Irina Grigorova, Barbara Baird, David Holowka, Kaushik Choudhuri, Erdinc Sezgin, Neetu Gupta, and Benjamin Machta for numerous helpful discussions. Research was funded by the NIH (R01GM110052) and the NSF (MCB 1552439).

## Additional information

### Funding

| Funder | Grant reference number | Author |
| --- | --- | --- |
| National Institute of General Medical Sciences | R01GM110052 | Sarah L Veatch |
| National Science Foundation | MCB 1552439 | Sarah L Veatch |

The funders had no role in study design, data collection and interpretation, or the decision to submit the work for publication.

### Author contributions

MBS, Conceptualization, Data curation, Software, Formal analysis, Validation, Investigation, Visualization, Methodology, Writing—original draft, Writing—review and editing; SAS, Conceptualization, Data curation, Software, Formal analysis, Supervision, Validation, Investigation, Visualization, Methodology, Writing—original draft, Writing—review and editing; MFN, Data curation, Formal analysis,

Investigation; KW, Investigation; SLV, Conceptualization, Resources, Data curation, Software, Formal analysis, Supervision, Funding acquisition, Validation, Investigation, Visualization, Methodology, Writing—original draft, Project administration, Writing—review and editing

### Author ORCIDs
Matthew B Stone, http://orcid.org/0000-0001-8858-4239
Sarah L Veatch, http://orcid.org/0000-0002-9317-2308

### Ethics
Animal experimentation: All experiments were performed in compliance with federal laws and institutional guidelines as approved by the University Committee on Use and Care of Animals (protocol #PRO00005048).

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
