## [Decision Letter]

Thank you for submitting your article "Protein sorting by phase-like domains supports emergent signaling function in B lymphocyte plasma membranes" for consideration by *eLife*. Your article has been favorably evaluated by Arup Chakraborty (Senior Editor) and three reviewers, one of whom, Michael L Dustin (Reviewer #1), is a member of our Board of Reviewing Editors. Your article has been reviewed by 3 peer reviewers, and the evaluation has been overseen by a Reviewing Editor and Arup Chakraborty as the Senior Editor.

The reviewers have discussed the reviews with one another and the Reviewing Editor has drafted this decision to help you prepare a revised submission.

Summary:

The authors have developed a set of mEos probes that show a preference for liquid ordered or liquid disordered lipid phases in giant plasma membrane vesicles. They have expressed these in a B cell in CH27 and allowed these cells to spontaneously adhere to glass surfaces. They perform super-resolution microscopy on the location of the probes in the presence or absence of BCR or cholera toxin binding site cross linking and establish very nice cross correlations between the organic dyes associated with the cross-linked surface receptors and the mEos probes. They find that the LO associated probes correlated with the BCR or CT clusters, whereas the TM, GG or CD45 TM based probes are anti correlated. They develop a model to explain BCR signalling in response to BCR cross linking in terms of phase separation of Src family kinases and CD45 at sites of BCR cross linking induced local lipid phase separation. They demonstrate that these effects are present in primary B cells and robust to the fixation taking place at 37 degrees.

Essential revisions:

1) The authors claim that these results represent the 'first direct visualization of both ordered and disordered domains in intact cells that resemble liquid-ordered and liquid-disordered phases in model membranes'. However, this does not appear to be the case as nearly all the findings presented here were published using fluorescence resonance energy transfer in living cells to follow the association of the BCR with membrane probes with time after BCR clustering (please see Sohn et al. 2006 PNAS; Sohn et al. 2008 J. Cell Biol.; and Tolar et al. 2005 Nat. Immunol.). In fact, the superresolution imaging is less well suited to the analysis as interactions of the BCRs with lipids are highly dynamic and as opposed to live cell imaging in which the association of the BCRs with lipids can be followed continuously the authors provide data for only two time points. For this manuscript to be considered for publication the authors would need to review the literature on this subject and make it clear what their novel contribution is.

Addressing this does not require new experiments, but a more explicit effort to put the work in context and clarify the novel aspects.

2) Making the case for partitioning into or out of lipid domains. In general, the work looks carefully performed. But as the authors admit, the spatial enrichment or depletion is subtle (20% quoted in the text). Thus, arguments validating the results could be stronger and the explanation improved. The results would be more convincing if validated using a structure that displays unambiguous co-localization on a 100 nm length scale.

A) Most of the images do not show what one would visually consider to be striking co-localization into the same domain and the "selected" boxes for magnification of this co-localization seem to be the minority of the field. The cross-relation function does not require precise overlap, but explores proximity, with correlations falling of at ~200 nm. Why they are not more precisely overlapped may require some explanation. For example, is this a consequence of photoactivation of limited number of fluorophores? How many fluorophores are being imaged in these "spots"? How did the authors define the time and number of fluorophores for activation?

B) Does the density of fluorophores affect the analysis? The image density looks quite variable. E.g. Figure 2. Can the authors do a test of various density to determine if this does not affect correlation functions?

C) It would be useful to validate the method and the scale with a control that one knows is precisely co-localized. A good one might be to transfect cells with two photoactivatable versions (different colors) of the clathrin HC which should precisely co-localize in coated pits.

D) It would be useful to show all of the dual color images in the supplement. For example, Figure 1 and Figure 2 show one pair of two or three that were tested in each panel. In Figure 3, no CD45 image is shown. Seems easy to show all images in a supplement.

E) The cell-to-cell variable looks considerable (Figure 1—figure supplement 3) with some cells not showing effects at all. Is this a concern for interpretation? I would show this cell-by-cell data for all experiments in the paper, not just Figure 1. It would also be good to categorize the variation in the cross-correlation function from cell to cell (e.g. a mean and variance between cells), rather than merge the data together. Also, it was not clear to me whether these were from cells on the same day or from completely different labeling and cell culture set ups. N =2 was described in the Methods, except for primary cells. I would add this info to the figure legends and also increase the N for the primary cells. Also in many cases the number of cells analyzed seems low, given that one can acquire a reasonable number super-resolution images of cells in one day (I would think more than 20 cells would be possible, especially as there are probably more than one cell in a field of view.).

F) It would be good to have some additional controls. As a computational control, can one randomize the positions of the clusters and then reanalyze? One would expect this to produce a flat correlation of 1. Probably true but would be nice to see. Also in Figure 1, it would be good to do imaging/analysis of the cholera toxin and peptides { ± } streptavidin (i.e. with and without streptavidin mediated clustering.

G) The authors comment that the co-clustering and exclusion effects are weak- on the order of 20%. However, this is not what the images look like- where the segregation appears to be 100%. The authors should present LUTs maps with the probe/receptor densities indicated against the color scale. Its assumed that this will explain the discrepancy. Some supplemental figures should be presented with the relevant signal presented in full range linear grey scale to show that "actual" surface densities with the actual level of co-clustering of segregation in BCR clusters indicated (perhaps the BCR cluster regions could be circles as they may not be as visible as in the contrast boosted versions. This will be helpful to general readers to understand why this issue has been so difficult to solve. If the CD45 exclusion is only <20% as suggested by the authors, then they should mention that biochemical analysis of 2D reaction from Hui et al., which they cite for different reasons, are consistent with quantitative effects of CD45 exclusion.

The most important new experiment suggested here is the control based on clathrin coated pits to demonstrate co-localization. The TM probe used by the authors may also be excluded from clathrin coated pits, although this will be a quantitative effect as it can be overcome by a clathrin pit localisation signal in the cytoplasm domain, but potentially also useful to validate that approach. The other points here are mostly related to analysis of existing data or better explanation, which should also be addressed as completely as possible.

2) Functional data

A) The main "functional" data is in Figure 4. However, these experiments are not tied into the earlier part of the paper to see if these cholesterol agents indeed alter correlated or anti-correlated distributions.

B) An effect on calcium signaling for the cholesterol agents is shown in supplement. Calcium signaling changes might result from general membrane changes and not BCR signaling specifically. Other markers of BCR signaling should be checked (e.g. phosphomarkers). Also, some parameter of "cell health" should be checked as well using these agents to make sure there are no drastic global effects.

C) Regarding the Cholera Toxin affect on Calcium signaling (although a more minor point in the paper). It looks like many cells are not changing their calcium levels. Can this cell to cell variability be checked? Also, is there a good control- e.g. another lipid binding protein or extracellular Fab antibody to ascertain whether this effect is really due to lipid clustering?

Experiments to demonstrate that the cholesterol altering treatments actually change the image correlations would be important to demonstrate a stronger correlation with function.

[Editors' note: further revisions were requested prior to acceptance, as described below.]

Thank you for resubmitting your work entitled "Protein sorting by lipid phase-like domains supports emergent signaling function in B lymphocyte plasma membranes" for further consideration at *eLife*. Your revised article has been favorably evaluated by Arup Chakraborty (Senior Editor), a Reviewing Editor, and two reviewers.

The manuscript has been improved but there are some remaining issues that need to be addressed before acceptance, as outlined below:

One of the reviewers had a remaining concern about the geranylgeranyl reporter. In the first figure you appear to discredit it as a reliable and accurate marker of lipid ordered phase-like domains (with an unsubstantiated explanation of electrostatic interactions). However, you then go on to treat it as a valid tool in subsequent figures, but again with complicated explanations. The GG reporter is usually a back-up for the TM reporter. Since the GG reporter failed in the CTxB tests in Figure 1 perhaps it is better to remove it and this would streamline the paper. The GG probe and variants of it could be the subject of a future study that dissects the acidic lipid hypothesis, which is interesting, but here just a distraction.

Subsection “Phase-like domains observed in intact B cells”, end of fifth paragraph: "…suggesting the probe expression impacts the mixing properties of the plasma membrane as a whole" This is not addressed further, but the two reviewers wonder how probe expression might be influencing many findings in the paper and this deserves some comment. One specific issue is that in most images, the PM, TM and CD45 probes appear to be in non-percolating, finite clusters and perhaps collectively cover <10% of the area. If both PM and TM probes appear to be in clusters, is there a percolating phase and which phase is this? The modelling certainly incorporates this concept, but how does the imaging, which usually shows isolated clusters relate to the modelled areas. Do the probes all appear clustered, because like the GG probe, they typically detect at least one other feature, in addition to the lipid order. Or is the appearance of clustering due to membrane topology?

The authors should show the real images for CD45 TM in Figure 3 and not in the supplement since this data is important for biological conclusions of the paper.

In the figure legends describing the simulation results, there should be an explicit description of the white and grey areas. Are there the disordered and ordered areas?

It would be useful to add the CD45 TM sequence to Figure 1—figure supplement 1 – just to show how it differs from the generic TM probe.

---

## [Author Response]

Essential revisions:

1) The authors claim that these results represent the 'first direct visualization of both ordered and disordered domains in intact cells that resemble liquid-ordered and liquid-disordered phases in model membranes'. However, this does not appear to be the case as nearly all the findings presented here were published using fluorescence resonance energy transfer in living cells to follow the association of the BCR with membrane probes with time after BCR clustering (please see Sohn et al. 2006 PNAS; Sohn et al. 2008 J. Cell Biol.; and Tolar et al. 2005 Nat. Immunol.). In fact, the superresolution imaging is less well suited to the analysis as interactions of the BCRs with lipids are highly dynamic and as opposed to live cell imaging in which the association of the BCRs with lipids can be followed continuously the authors provide data for only two time points. For this manuscript to be considered for publication the authors would need to review the literature on this subject and make it clear what their novel contribution is.

Addressing this does not require new experiments, but a more explicit effort to put the work in context and clarify the novel aspects.

We thank the reviewer for this comment and we have made changes to the text, mainly in the Introduction and conclusions sections, in an effort to better place our work in the context of existing literature and define our novel contribution. Specifically, in the Introduction we describe in more detail how FRET has been used to investigate membrane heterogeneity in general and briefly discuss a few relevant drawbacks of FRET techniques:

“Some of the strongest experimental evidence supporting a heterogeneous plasma membrane comes from Förster resonance energy transfer (FRET) measurements between membrane components. […] This approach allows us to characterize and quantify, in a model-independent manner, the spatial organization of membrane components on length scales between those accessible by FRET-based techniques and conventional optical microscopy.”

We also describe the previous evidence from FRET experiments that association with ordered lipids is important for BCR signaling in particular:

“A third model, which is the focus of investigation here, postulates that clustering BCR acts to stabilize an ordered membrane domain that impacts the receptor-proximal distribution of regulatory proteins involved in initiating or modulating the resulting cellular response. […] Currently, the role of membrane domains in BCR signaling remains a topic of active investigation (Pierce and Liu 2010; Horejsi and Hrdinka 2014) as the field works to put together a more holistic picture of how interactions between proteins and lipids could support signaling function.”

We place this theory in the context of other theories of the mechanism of BCR activation in this paragraph of the Introduction. In the last paragraph of the Introduction, we have also expanded upon the description of our novel contribution and have added substantial text to the Conclusions section that addresses this point.

We now also include citations of Sohn et al. 2006 and 2008 in the discussion of Figure 2 in the Results and Discussion and mention that although transient association between BCR and ordered lipid markers was observed in these FRET measurements, BCR cross-correlations with the PM lipid probe appear to be sustained at least over the 1 to 5 min time window that we probe in these experiments:

“The direct measurements of peptide sorting shown here are generally consistent with past FRET and biochemical isolation measurements that argued that clustered BCR resides within ordered membrane domains (Cheng et al. 1999; Pierce 2002; Sohn et al. 2006; Sohn et al. 2008b). The association between clustered BCR and the ordered domain marker PM appears more sustained in our imaging measurements than was observed in past reports using FRET (Sohn et al. 2006; Sohn et al. 2008b), possibly due to the different length-scales probed by these methods.”

2) Making the case for partitioning into or out of lipid domains. In general, the work looks carefully performed. But as the authors admit, the spatial enrichment or depletion is subtle (20% quoted in the text). Thus, arguments validating the results could be stronger and the explanation improved. The results would be more convincing if validated using a structure that displays unambiguous co-localization on a 100 nm length scale.

The revised manuscript contains a much more extensive discussion of our methods and analysis of the error present in our measurements, and we hope that this will aid readers in interpreting the data. To this end, content has been added to the Materials and methods section including a new Figure 6, and we have also added numerous figure supplements, specified below. As an additional validation of our imaging and analysis methods, Figure 6 includes data from a positive control experiment suggested by reviewers where clathrin was labeled in two color channels by double-transfection with two different clathrin heavy chain fluorescent fusion constructs. This exogenous fluorescent clathrin is incorporated into the same clathrin-coated pit structures and is clearly co-localized in super-resolution images and according to our cross-correlation analysis.

A) Most of the images do not show what one would visually consider to be striking co-localization into the same domain and the "selected" boxes for magnification of this co-localization seem to be the minority of the field. The cross-relation function does not require precise overlap, but explores proximity, with correlations falling of at ~200 nm. Why they are not more precisely overlapped may require some explanation. For example, is this a consequence of photoactivation of limited number of fluorophores? How many fluorophores are being imaged in these "spots"? How did the authors define the time and number of fluorophores for activation?

We agree with the reviewer that a visual interpretation of the super-resolution images can seem to contradict the results of the cross-correlation analysis in some cases, but we feel that this is primarily due to the inherent difficulty in interpreting these poorly sampled pointillistic images by eye. In our case, many of the clusters lack of apparent overlap with membrane probes because the underlying distributions of the membrane probe do not sample all space. That is, the labeling density of the membrane probe is sparse compared to the spatial resolution of the measurement. This is mainly due to limits of expression of this exogenous probe. In conventional fluorescence microscopy images, unless labeling density is very low, typically structures are sampled at a density higher than the resolution of the measurement. In other words, labeling density is higher than 1/(250nm)^2 and the labeling of structures appears continuous. This is often not the case for super-resolution microscopy images because the lateral resolution is considerably better. As a result, especially if labels are more or less randomly distributed across the membrane as is the case for the lipid probes, the images give the visual impression that the labels are self-clustered in domains. In fact, individual labels, which are over-counted, are separated from each other by distances greater than the resolution of the measurement but their distribution is still close to random. This can be more easily seen if fixed-cell super resolution images are compared to live cell super-resolution images (shown in Figure 2 and Figure 2—figure supplement 11). The distributions of lipid probes appear more uniform in live cell measurements because probes are mobile and allowed to sample more lateral space as they reversibly blink on and off, but the cross-correlation curves show nearly the same relative distributions between BCR and lipid probes as in fixed cells. To help illustrate this idea we have also included a figure supplement to Figure 6 (Figure 6—figure supplement 3), which shows that decreasing probe density impacts the visual impression of apparent co-localization as well as statistical variation in the correlation function, but does not change the average cross-correlation function.

The width of the cross-correlation mentioned by the reviewers is determined by the size of the correlated structures, in this case protein clusters and associated membrane domains. The fall-off of the cross-correlation function is gradual because it is an average measurement of all clusters over multiple cells and there can be a range of cluster sizes sampled, and also because the structures are blurred by the limited resolution of the measurement.

B) Does the density of fluorophores affect the analysis? The image density looks quite variable. E.g. Figure 2. Can the authors do a test of various density to determine if this does not affect correlation functions?

As mentioned above, the apparent difference in the density of PM labels in Figure 2 vs. 2C in particular is mostly due to the fact that the cell shown in Figure 2 was imaged live and the cell shown in Figure 2 was imaged fixed. In both cases individual mEos3.2 labels are localized multiple times because the probe blinks reversibly. In the case of the live cell, PM-mEos3.2 labels diffuse around the cell footprint as they are imaged, and also exchange between the top and bottom surfaces of the cell. As a result a greater fraction of the cell footprint in “filled in” by PM-mEos3.2 localizations, leading to the impression of a more uniform distribution of mEos3.2 and higher density.

However, the reviewer is correct that there was a range of labeling densities for the lipid probe in these images due to variation in the expression level of the transfected probe. We find that there are subtle correlations between lipid probe density and the value of the cross-correlation and present this analysis in figure supplements Figure 1—figure supplement 6 and Figure 2—figure supplement 3.

C) It would be useful to validate the method and the scale with a control that one knows is precisely co-localized. A good one might be to transfect cells with two photoactivatable versions (different colors) of the clathrin HC which should precisely co-localize in coated pits.

We thank the reviewers for this suggestion and think that inclusion of this positive control has improved the clarity of the manuscript. We performed this experiment by co-transfecting HeLa cells with clathrin heavy chain fluorescent constructs and show the super-resolution data and analysis in Figure 6. These two orthogonally labeled populations of clathrin are highly cross-correlated by our analysis. STORM images obtained from this data appear highly colocalized in clathrin pits of characteristic size, as expected from previous work. This control data is discussed in detail in the Materials and methods section under the heading “Cross-correlation analysis in chemically fixed cells”.

D) It would be useful to show all of the dual color images in the supplement. For example, Figure 1 and Figure 2 show one pair of two or three that were tested in each panel. In Figure 3, no CD45 image is shown. Seems easy to show all images in a supplement.

We have added figure supplements that show representative super-resolution images for conditions included in cross-correlation curves but not shown in the main figures. For example, Figure 1—figure supplement 7 shows representative images for CTxB clusters with GG, CTxB clusters with TM, and TM clusters with GG. New figure supplements include Figure 1—figure supplement 7, Figure 2—figure supplement 11, and Figure 3—figure supplement 4.

E) The cell-to-cell variable looks considerable (Figure 1—figure supplement 3) with some cells not showing effects at all. Is this a concern for interpretation? I would show this cell-by-cell data for all experiments in the paper, not just Figure 1. It would also be good to categorize the variation in the cross-correlation function from cell to cell (e.g. a mean and variance between cells), rather than merge the data together. Also, it was not clear to me whether these were from cells on the same day or from completely different labeling and cell culture set ups. N =2 was described in the Methods, except for primary cells. I would add this info to the figure legends and also increase the N for the primary cells. Also in many cases the number of cells analyzed seems low, given that one can acquire a reasonable number super-resolution images of cells in one day (I would think more than 20 cells would be possible, especially as there are probably more than one cell in a field of view.).

To clarify the sample sizes analyzed in imaging experiments, we now state the number of cells included in average cross-correlation curves in the corresponding figure captions and figure supplements. We have also modified the statement in the Materials and methods regarding biological replicates:

“The number of cells going into each average is shown in figure captions or figure supplements and, with the exception of the primary cell experiment, includes at least two biological replicates where samples were prepared on separate days.”

We have also characterized the cell-to-cell variation in the cross-correlation function as well as the sources of error present in individual cross-correlation functions much more completely than in the previous version of the manuscript. First, we have developed a better estimate of the variance of an individual cross-correlation measurement derived from a single cell by estimating the total number of independent objects (labeled proteins of interest) in our data using the auto-correlation function. The methodology used for this estimate is described in detail in new additions to the Materials and methods under the headings “Over-counting and estimating protein/peptide surface densities” and “Variance of cross-correlation measurements”, and is shown graphically in Figure 6—figure supplement 5.

To better capture the cell-to-cell variation present in the imaging data, we have also included several new figure supplements. In Figure 1—figure supplement 5 and Figure 2—figure supplement 2, we plot distributions of C(r<25nm) values from individual cells that are included in the CTxB clustering and BCR clustering experiments shown in Figure 1 and 2. We show that the width of these distributions is well matched by the average of the variances expected from individual measurements. This indicates that the bulk of the variation within average cross-correlations can be attributed to variance due to underlying sampling and number statistics present in individual measurements, rather than cell-to-cell variation. Finally, we now show cross-correlation functions from individual cells for both the BCR cluster data in Figure 2—figure supplement 1 as well as the CTxB cluster data in Figure 1—figure supplement 4. We feel that these efforts add significantly to the quality of this manuscript.

In its current form, the manuscript describes 41 distinct super-resolution imaging experiments. For cases where the statistics or other experimental issues warranted (see above), up to 50 cells were imaged for a given condition. This level of scrutiny was not required for all measurements, which is why some observations were probed in ~10 cells. Also, because we have limited access to and expertise in performing experiments involving the use of animals, we were not able to easily repeat the primary cell experiment.

F) It would be good to have some additional controls. As a computational control, can one randomize the positions of the clusters and then reanalyze? One would expect this to produce a flat correlation of 1. Probably true but would be nice to see. Also in Figure 1, it would be good to do imaging/analysis of the cholera toxin and peptides { ± } streptavidin (i.e. with and without streptavidin mediated clustering.

We thank the reviewer for this comment and have incorporated additional computational controls as suggested in a supplement to Figure 6, Figure 6—figure supplement 1. In this figure we show simulated data and resulting cross-correlation curves when the molecules in each channel have various distributions, and the two channels also have various relative distributions. Here, we show that whenever the distribution of probes in one channel is random with respect to the other channel, the cross-correlation is flat, equal to 1. This occurs regardless of whether the probes in one or both channels are randomly distributed or clustered. When probes in both channels are co-clustered, however, the cross-correlation function reflects this co-localization.

As mentioned above, we have now included imaging data where unclustered CTxB is imaged in conjunction with PM, TM, and GG peptides in Figure 2—figure supplement 10. We briefly discuss these results in the context of unclustered BCR data in the Results and Discussion section:

“We additionally observed weak PM enrichment and GG depletion around unclusterd BCR in chemically fixed CH27 cells, although the magnitude of this sorting is on the edge of the sensitivity limits of our imaging system (Figure 2—figure supplement 10). […] Improved lateral resolution is needed to systematically investigate the lateral organization of receptors and peptides in intact cells without receptor clustering, and may be enabled by recent improvements in fluorophores and imaging modalities (Huang et al. 2013; Grimm et al. 2015; Grimm et al. 2016).”

G) The authors comment that the co-clustering and exclusion effects are weak- on the order of 20%. However, this is not what the images look like- where the segregation appears to be 100%. The authors should present LUTs maps with the probe/receptor densities indicated against the color scale. Its assumed that this will explain the discrepancy. Some supplemental figures should be presented with the relevant signal presented in full range linear grey scale to show that "actual" surface densities with the actual level of co-clustering of segregation in BCR clusters indicated (perhaps the BCR cluster regions could be circles as they may not be as visible as in the contrast boosted versions. This will be helpful to general readers to understand why this issue has been so difficult to solve.

We thank the reviewer for this comment, and we have included colorbars in Figure 6—figure supplement 2 in order to show the number of raw localizations in the image more plainly. We would also like to point out that the number of localizations of fluorophore tagging the BCR does not correspond to the density of BCR, rather the number of localizations will scale with the experimental parameters including choices of buffer, number of frames observed, and specific fluorophore used. Here, we estimate the overall density of the unclustered mEos3.2 probe using image autocorrelation analysis, however, this analysis provides an underestimate of the BCR-Atto655 density due to the clustering of BCR. This is now discussed explicitly in a new section of Materials and methods under the heading “Over-counting and estimating protein/peptide surface densities”. As explained in response to A) above, it can be difficult to judge the extent of co-localization in these images by eye because the distributions of proteins of interest under-sample space. This is also the reason why even distributions of probes that are not clustered appear punctate.

If the CD45 exclusion is only <20% as suggested by the authors, then they should mention that biochemical analysis of 2D reaction from Hui et al., which they cite for different reasons, are consistent with quantitative effects of CD45 exclusion.

This is an interesting point and a good opportunity to discuss our results in the context of concentration differences that might significantly impact signaling. Hui et al. (Nat. Struct. and Molec. Bio. 2014) showed that, in a reconstituted system, that the phosphorylation state of the CD3ζ subunit of the T cell receptor is “ultrasensitive” to changes in the relative concentrations of Lck and CD45 and displays switch-like behavior in the transition from the inactive to active states at physiological concentrations of both enzymes. They also showed that perturbations that alter the spatial organization of CD3ζ and/or Lck such as receptor clustering could alter the “switch” concentrations at which CD3ζ becomes phosphorylated. We observe not only a depletion of CD45 but also an enrichment of Lyn co-localized with BCR clusters, and if we observe a 20% increase in Lyn due to its lipid anchor and a 20% decrease in CD45 concentrations within BCR domains, this amounts to a 50% change overall in the ratio of Lyn to CD45 attributable to lipid-driven sorting. If we draw a direct analogy with the cooperativity observed in the Lck/CD45/CD3ζ system, a 50% change in Lck/CD45 ratio could at most increase the total fraction of activated BCR by 20%. This difference very well may be sufficient for surpassing some signaling threshold for cell activation, and additional mechanisms of positive feedback (e.g. local regulation of Lyn activity) could amplify the effect.

We now discuss our results in the context of the Hui et al. paper in the Results and Discussion section:

“It is reasonable to expect that the ~50% increase in the ratio of Lyn to CD45 due to sorting by ordered domains could cause a significant change in BCR phosphorylation levels if we compare to results obtained for the related T cell receptor (TCR) system. […] Additionally, the actual enrichment and depletion of probes around BCR and CTxB is larger than the measured values presented here since the real spatial distributions are convolved with the finite resolution of the measurement to give the observed cross-correlations.”

The most important new experiment suggested here is the control based on clathrin coated pits to demonstrate co-localization. The TM probe used by the authors may also be excluded from clathrin coated pits, although this will be a quantitative effect as it can be overcome by a clathrin pit localisation signal in the cytoplasm domain, but potentially also useful to validate that approach. The other points here are mostly related to analysis of existing data or better explanation, which should also be addressed as completely as possible.

As explained in C) above, we performed the 2-color clathrin control experiment and now show this data both as a validation of our imaging and analysis methods and as a demonstrative example to better explain the method for readers who may not be familiar with super-resolution techniques.

We hope that the addition of the clathrin control experiment, additions of new data, simulations, and discussion in response to the specific comments above, and our efforts to clarify data analysis methodology with the inclusion of the new Figure 6, as well as substantial expansion of the Materials and methods will sufficiently validate our approach in the eyes of the reviewers.

2) Functional data

A) The main "functional" data is in Figure 4. However, these experiments are not tied into the earlier part of the paper to see if these cholesterol agents indeed alter correlated or anti-correlated distributions.

We agree that showing the effects of cholesterol perturbations on BCR/lipid probe cross-correlations would greatly strengthen the connection between the predictions of the model and the effects of cholesterol perturbation on Ca^2+^ mobilization in B cells, and we thank the reviewer for encouraging us to conduct this measurement. We performed super resolution imaging experiments where CH27 cells were subjected to cholesterol perturbations and the cross-correlation between clustered BCR and the PM lipid probe was measured. These data are now shown in Figure 5 of the revised manuscript. Our results indicate that cholesterol addition lowers the cross-correlation between BCR and PM, whereas cholesterol removal increases the cross-correlation between BCR and PM. These experimental results are in qualitative agreement with cross-correlation curves generated from simulations where the surface fraction of ordered vs. disordered components was varied, shown in Figure 5 of the revised manuscript.

B) An effect on calcium signaling for the cholesterol agents is shown in supplement. Calcium signaling changes might result from general membrane changes and not BCR signaling specifically. Other markers of BCR signaling should be checked (e.g. phosphomarkers). Also, some parameter of "cell health" should be checked as well using these agents to make sure there are no drastic global effects.

We performed viability experiments on cells treated with cholesterol agents to ensure that the effects on Ca^2+^ mobilization that we observed are not attributable to destabilization of the cell membrane alone. We subjected CH27 cells to the same cholesterol perturbations that were used in Ca^2+^ mobilization experiments and used Annexin V labeling of the outer membrane as a marker for PS exposure that occurs during apoptosis. We find no significant differences in membrane binding of Annexin V following treatment of cells with cholesterol agents, indicating that the viability of these cells is not compromised by the cholesterol perturbations used for experiments. These data are now shown in Figure 5—figure supplement 3 of the revised manuscript. We also conducted western blots using the 4G10 anti-pY antibody in cells with cholesterol perturbations as a second functional assay of BCR signaling. In our hands, these measurements were highly variable and a meaningful trend did not emerge in this large background, therefore were not included in the revised manuscript.

C) Regarding the Cholera Toxin affect on Calcium signaling (although a more minor point in the paper). It looks like many cells are not changing their calcium levels. Can this cell to cell variability be checked? Also, is there a good control- e.g. another lipid binding protein or extracellular Fab antibody to ascertain whether this effect is really due to lipid clustering?

We have altered this figure (Figure 3—figure supplement 1 of the revised manuscript) in order to show the standard deviation of the Ca^2+^ response of individual cells following CTxB clustering in addition to the average Ca^2+^ response. Instead of plotting all of the individual cell traces, which may have obscured the dynamics of individual cells, we also chose to plot a few selected Ca^2+^ traces that represent the variation in individual cell responses. To further explore the signaling response in B cells initiated by CTxB, we have also performed western blotting experiments that probed protein phosphorylation following CTxB binding and clustering. These data are shown in Figure 3—figure supplement 3 of the revised manuscript and show that several of the same targets that are phosphorylated as a result of BCR stimulation, including BCR itself, Lyn, and Syk, are also phosphorylated as a result of CTxB clustering. This complements the data presented in Figure 3, which shows that ordered domains induced by CTxB clusters sort BCR signaling partners and provide an environment that promotes protein phosphorylation, including activation of Lyn kinase. It also complements the new simulations added to Figure 4 showing that formation of an ordered domain alone is sufficient to cause BCR activation. We feel that the new data and simulations strengthen our conclusion that CTxB clustering, through formation of an ordered domain, creates a local environment that favors BCR activation.

Experiments to demonstrate that the cholesterol altering treatments actually change the image correlations would be important to demonstrate a stronger correlation with function.

We thank the reviewers for this feedback and have performed the experiments suggested, as detailed in A), above. We have also performed additional functional experiments to expand on existing data, as well as controls testing cell viability. We hope that these changes clarify and strengthen the arguments made in the revised manuscript.

[Editors' note: further revisions were requested prior to acceptance, as described below.]

The manuscript has been improved but there are some remaining issues that need to be addressed before acceptance, as outlined below:

One of the reviewers had a remaining concern about the geranylgeranyl reporter. In the first figure you appear to discredit it as a reliable and accurate marker of lipid ordered phase-like domains (with an unsubstantiated explanation of electrostatic interactions). However, you then go on to treat it as a valid tool in subsequent figures, but again with complicated explanations. The GG reporter is usually a back-up for the TM reporter. Since the GG reporter failed in the CTxB tests in Figure 1 perhaps it is better to remove it and this would streamline the paper. The GG probe and variants of it could be the subject of a future study that dissects the acidic lipid hypothesis, which is interesting, but here just a distraction.

We agree that the inclusion of the GG data complicates the story without substantially adding to the manuscript. As suggested, we have removed these results from the manuscript except for Figure 1. Here, we feel it adds by showing recruitment of a disordered marker to the TM clusters, showing that lipid probe recruitment to disordered clusters is symmetrical but opposite to recruitment of lipid probes to ordered clusters. Unlike BCR or CTxB clusters, we also have no basis to hypothesize that charged lipids are recruited to TM clusters. We do plan on investigating the electrostatic interactions of this probe further and agree that additional experimentation would be required to support the acidic lipid hypothesis surrounding the GG results posed in the previous version.

Removal of the GG data resulted in modification of Figure 1 and Figure 2 and related supplemental figures, including removal of Figure 2—figure supplement 4 of the previous version of the manuscript, which included cross-correlation functions of BCR with a PH domain. We have also removed two paragraphs from the Results section that provided possible electrostatics-related explanations and context for the GG results. No other substantive changes were made related to the removal of GG data.

Subsection “Phase-like domains observed in intact B cells”, end of fifth paragraph: "…suggesting the probe expression impacts the mixing properties of the plasma membrane as a whole" This is not addressed further, but the two reviewers wonder how probe expression might be influencing many findings in the paper and this deserves some comment.

We have now added additional sentences in the text surrounding the discussion of Figure 1 and Figure 2 to directly address this issue:

“Both the expression-level dependence of probe partitioning and the small size of CTxB and TM clusters impede us from drawing quantitative conclusions regarding the magnitude of enrichment or depletion of peptides into domains. Instead, we draw conclusions regarding whether probes are enriched or depleted within our sensitivity limits which is not impacted by either of these factors.”

“Again, we find that the expression level of phase sensitive peptides impacts the magnitude but not the sign of peptide partitioning with respect to clustered BCR, allowing us to determine the type of domain stabilized by BCR clusters if not its quantitative composition.”

One specific issue is that in most images, the PM, TM and CD45 probes appear to be in non-percolating, finite clusters and perhaps collectively cover <10% of the area. If both PM and TM probes appear to be in clusters, is there a percolating phase and which phase is this? The modelling certainly incorporates this concept, but how does the imaging, which usually shows isolated clusters relate to the modelled areas. Do the probes all appear clustered, because like the GG probe, they typically detect at least one other feature, in addition to the lipid order. Or is the appearance of clustering due to membrane topology?

Thank you for this question that points out a way in which the manuscript was unclear. We have now expanded Figure 1—figure supplement 3 and included a few more comments in the main text to address these issues. We have also recently written a review article that deeply discusses complications that arise in interpreting super-resolution images of membranes including several factors relevant to the points made above. This review article was recently accepted and is now referenced in the revised manuscript.

First, we cannot comment directly on percolation because our super-resolved images are inherently under-sampled in space. Briefly, the apparent amount of surface area taken up by a peptide probe in super-resolution images is given by its expression level and the resolution of the measurement. This apparent surface fraction does not directly inform the surface fraction taken up by the entire domain or underlying membrane structure that is labeled by the peptide. This is because the labeled peptide is just one of many components that make up the domain, and the average distance between labeled peptides is much larger than the localization precision of the measurement. We now show how this inherent under-sampling of space impacts the visual impression given by reconstructed images and the resulting cross-correlation function within Figure 1—figure supplement 3. We also added the following text near the beginning of the Results section:

“We note that the apparent surface area occupied by peptides is likely a vast under-estimate of the surface area occupied by ordered or disordered phase-like domains since peptides are only one of many membrane components that likely occupy these domains, leading to inherent under-sampling of space (Figure 1—figure supplement 3).”

Second, the apparent self-clustering of probes is most likely due to multiple observations of the same fluorophore (over-counting) and not due to any inherent self-clustered structure, with the clear exception of situations where proteins are intentionally clustered such BCR clustered by streptavidin. In these cases where clustering is not enforced, a detailed statistical analysis of our images indicates that each independent labeled protein or peptide is sampled at least 10 times (on average) in a given image, allowing the super-resolved point spread function to be well sampled. While we cannot exclude the possibility that labeled proteins or peptides are not independently distributed due to some strong self-clustering interaction, it is more likely that the appearance of self-clustering comes from multiple observations of the same probe, since we are not attempting to limit the number of times each protein or peptide is observed. In fact, we can be confident that we are observing single BCRs multiple times since we are using a fAb that can bind to multiple sites within single receptors, and multiple fluorophores bind to single fAbs. Also, the fluorophore conjugated to the peptide constructs (mEos3.2) is known to reversibly photo-activate under our imaging conditions, leading to over-counting of mEos3.2 labeled peptides.

We have added additional text to address this issue early in the main text of the manuscript.

“In addition, incomplete spatial sampling by fluorescent peptides can give rise to the appearance of peptide self-clustering, since a single mEos3.2 fluorophore can reversibly photo-switch (Endesfelder et al. 2011) and therefore is likely detected multiple times over the course of a measurement (Stone et al. In Press; Veatch et al. 2012).” This concept is addressed in greater detail in Methods and in several figure supplements accompanying Figure 6. As stated above, the manuscript now cites a recently accepted review article by three of the authors that cover these (and other) topics in greater depth.

The authors should show the real images for CD45 TM in Figure 3 and not in the supplement since this data is important for biological conclusions of the paper.

We have now added images of BCR/CD45 TM and CTxB CD45 to Figure 3.

In the figure legends describing the simulation results, there should be an explicit description of the white and grey areas. Are there the disordered and ordered areas?

We have now added this information to the legend in Figure 4.

It would be useful to add the CD45 TM sequence to Figure 1—figure supplement 1 – just to show how it differs from the generic TM probe.

We have now added this to Figure 1—figure supplement 1 along with the other peptide constructs used in this study.